

# On the matter of topological insulators as magnetoelectric

**N. P. Armitage[1*] and Liang Wu[2]**

**1** The Institute for Quantum Matter, Department of Physics and Astronomy,
The Johns Hopkins University, Baltimore, MD 21218 USA.
**2** Department of Physics and Astronomy, University of Pennsylvania,
Philadelphia, PA 19104 USA.

* npa@pha.jhu.edu

## Abstract

It has been proposed that topological insulators can be best characterized not as *surface* conductors, but as *bulk magnetoelectrics* that – under the right conditions– have a universal quantized magnetoelectric response coefficient $e^2/2h$. However, it is not clear to what extent these conditions are achievable in real materials that can have disorder, finite chemical potential, residual dissipation, and even inversion symmetry. This has led to some confusion and misconceptions. The primary goal of this work is to illustrate exactly under what real life scenarios and in what context topological insulators can be described as magnetoelectrics. We explore analogies of the 3D magnetoelectric response to electric polarization in 1D in detail, the formal vs. effective polarization and magnetoelectric susceptibility, the $\frac{1}{2}$ quantized surface quantum Hall effect, the multivalued nature of the magnetoelectric susceptibility, the role of inversion symmetry, the effects of dissipation, and the necessity for finite frequency measurements. We present these issues from the perspective of experimentalists who have struggled to take the beautiful theoretical ideas and to try to measure their (sometimes subtle) physical consequences in messy real material systems.



# 1   Introduction

Topological insulators (TI) are a recently discovered class of materials that are in the ideal (e.g. in the absence of bulk conductivity) characterized as bulk insulators with topologically protected surface states [1–3]. Although in many cases it is sufficient to characterize them as *surface* conductors it has been proposed that topological insulators are – with some considerations – better characterized as *bulk* magnetoelectrics [4, 5]. Indeed we will see that to understand some of their aspects this perspective is essential. However this perspective has led to some confusion and misconceptions. The goal of this work is to give some insight into how one can regard TIs as magnetoelectrics and how this can give a more complete characterization of their properties. These issues are important because this quantized magnetoelectric response is the 2nd example (with the quantum Hall effect the first) where a topological quantum number can in principle be measured directly via a response function.

A number of aspects are highlighted here. We explore in detail analogies of the 3D magnetoelectric response to electric polarization in 1D. The *formal* polarization of a bulk sample is a multivalued quantity in contrast to the single-valued *effective* polarization of actual crystallite. We can then make a direct analogy to the multivalued formal magnetoelectric susceptibility of a bulk magnetoelectric vs. the single valued effective magnetoelectric susceptibility. This analogy also leads to insight regarding the role of inversion symmetry in these topological systems and demonstrates how the $\frac{1}{2}$ quantized surface quantum Hall effect of an inversion symmetric topological insulator arises as a higher dimensional analog of the $\frac{1}{2}$ quantized end charges of a 1D inversion symmetric chain. Moreover, in just the same fashion as the effective polarization can only be defined in a charge neutral system, the effective magnetoelectric susceptibility can only be defined in a system whose net Hall response is zero. However the formal polarization and magnetoelectric susceptibility can be defined independent of these considerations. In fact, measurement of a single end charge or one surface Hall conductance is sufficient to establish the formal polarization and magnetoelectric susceptibility. Further insight is gained by making analogies regarding Thouless pumps in both cases as well. Finally, we show that a "true" effective magnetoelectric response e.g. a dc electric polarization being created by a

dc applied magnetic field or a dc magnetization being created by a dc applied electric field can only occur in a topological insulator under a *very* restricted set of material conditions. These conditions are essentially unrealizable with current (and perhaps foreseeable) material considerations. However, under conditions fulfilled in real experiments, the ac response at very low frequencies exhibits a response indistinguishable from a magnetoelectric and in this regard, it is appropriate to characterize real topological insulators as magnetoelectrics.

Magnetoelectrics are materials in which an electric polarization can be created by an applied magnetic field or a magnetization can be created by an applied electric field. They have been topics of interest for decades [6–10]. Representative examples of magnetoelectric (ME) materials are $Cr_2O_3$ [8], which has an ME coupling with a $\mathbf{E} \cdot \mathbf{B}$ ME coupling and multiferroic $BiFeO_3$ [11] which has a ME coupling that can be written (in part) in a $\mathbf{E} \times \mathbf{B}$ form. The linear magnetoelectric tensor is defined as

$$\alpha_{ij} = \left.\frac{\partial P_i}{\partial B_j}\right|_{\mathbf{E} \to 0} = \left.\frac{\partial M_i}{\partial E_j}\right|_{\mathbf{B} \to 0}. \tag{1}$$

In general this response contains both "frozen-ion" and "lattice-mediated" contributions. Each of these can be further separated into spin and orbital parts. It has been proposed that topological insulators are best characterized not as surface conductors, but as special $\mathbf{E} \cdot \mathbf{B}$ magnetoelectrics [4,5] with a frozen-ion orbital response that gives a diagonal and uniform contribution to Eq. 1 and whose size is quantized to be half-integer multiples of the fundamental von Klitzing constant $e^2/h$ e.g $\alpha = (N + \frac{1}{2})\frac{e^2}{h}$ (where $N$ is an integer). In the topological field theory this can be shown to be a consequence of an additional term,

$$\mathcal{L}_\theta = 2\alpha\sqrt{\frac{\epsilon_0}{\mu_0}}\frac{\theta}{2\pi}\mathbf{E} \cdot \mathbf{B}, \tag{2}$$

added to the usual Maxwell Lagrangian [4]. Here $\epsilon_0$ and $\mu_0$ are the permittivity and permeability of free space. $\theta$ is the "axion angle" that will be defined in more detail below, but in a material that has either time reversal ($\mathcal{T}$) or inversion ($\mathcal{P}$) symmetry it is constrained to be an integer times $\pi$. In topological insulators it is an odd multiple of $\pi$ and in trivial insulators, it is an even integer times $\pi$. This defines the "strong" $Z_2$ topological index that assumes values of either 1 or 0. As we will see below, defining these systems as bulk magnetoelectrics has the advantage of not only allowing explanation of the quantized ME response, but is also more in keeping with how we usually define response functions of homogeneous materials, as we can describe the physics in terms of a bulk response function without making explicit reference to surface states.

An analogy can be made between the physics described by $\mathcal{L}_\theta$ to that of the hypothetical field/particle that was proposed by Peccei and Quinn, Weinberg, and Wilczek to explain the small charge conjugation parity (CP) symmetry violation in the strong interaction (for instance the lack of a large neutron electric dipole moment) [12–15]. Wilczek called the particle the *axion* after a brand of laundry detergent (Fig. 1) because they "cleaned up" a problem with CP violation [16]. The fundamental axion particle has not been observed in particle physics experiments, but one may study a related effect in the context of magnetoelectrics [15,17]. For these reasons the topological magnetoelectric effect (TME) of the kind that appears in TIs has been called "axion electrodynamics".

Although $\mathcal{L}_\theta$ is generic expression which can be applied for instance to $Cr_2O_3$ (with a $\theta \ll \pi$ [17–20]) or in a astrophysical context [21–23] its form merits additional discussion when applied to TIs. Moreover, there are a number of aspects that require clarification in regarding TIs as magnetoelectrics. For instance, it is usually taken as a given that one must break both $\mathcal{T}$ and $\mathcal{P}$ to have a finite magnetoelectric coefficient. Indeed this was part of Dzyaloshinskii's original considerations [8]. However, many TI materials (even if $\mathcal{T}$ is broken

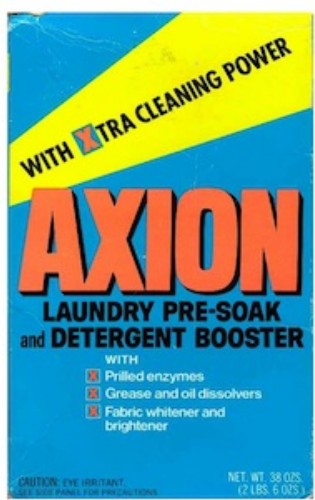

Figure 1: Axion laundry detergent was Wilczek's inspiration for the name of his particle that "cleaned up" a problem with CP violation. Wilczek has written, "I called this particle the axion, after the laundry detergent, because that was a nice catchy name that sounded like a particle and because this particular particle solved a problem involving axial currents." [16]

by an applied magnetic field or inherent magnetism) do not break inversion in their bulk. How then can a TI be characterized as a magnetoelectric? Indeed these issues have led to much confusion and debate [24]. Moreover, it is not clear to what extent the considerations of the beautiful field theoretic formulations hold up in real materials. For instance, what is the role of dissipation and disorder at the surfaces? In what circumstances is it better to regard TIs as bulk magnetoelectrics vs. surface conductors? We will address these issues and others in this work.

With the possible exception of the explicit formulation of the formal magnetoelectric susceptibility as a multivalued lattice and the discussion on the role of dissipative effects, there is very little that is truly new in this manuscript. And although some parts of it should be considered very elementary, we hope that this manuscript's sometimes unconventional presentation means that even experts working in the area of topological materials will find it novel, interesting and useful. Many people who have understand these issues may have understood them only in a quite different context or considered aspects too elementary to write down explicitly. Others may have found existing treatments too opaque. Even casual perusal of this manuscript should make clear that our goal here is not rigor. We present these issues from the perspective of experimentalists who have struggled to take beautiful theoretical ideas and to try to measure their (sometimes subtle) physical consequences in messy real material systems. More rigorous and advanced treatment of some of these concepts can be found in original literature and a number of excellent reviews [25–33].

## 2  "Modified" Maxwell's equations for the ideal case

As mentioned above, Qi et al. [4] showed that the electrodynamics of topological insulators can be described by adding a topological term $\mathcal{L}_\theta = 2\alpha \sqrt{\frac{\epsilon_0}{\mu_0}} \frac{\theta}{2\pi} \mathbf{E} \cdot \mathbf{B}$ to the usual Maxwell Lagrangian $\mathcal{L}_0$. The consequences of this additional term gives modified Gauss's and Ampère's law with

new source and current contributions that read

$$\nabla \cdot \mathbf{E} = \frac{\rho}{\epsilon_0} - 2c\alpha\nabla\left(\frac{\theta}{2\pi}\right)\cdot\mathbf{B}, \tag{3}$$

$$\nabla \times \mathbf{B} = \mu_0\mathbf{J} + \frac{1}{c^2}\frac{\partial\mathbf{E}}{\partial t} + \frac{2\alpha}{c}\left[\mathbf{B}\frac{\partial}{\partial t}\left(\frac{\theta}{2\pi}\right) + \nabla\left(\frac{\theta}{2\pi}\right)\times\mathbf{E}\right]. \tag{4}$$

In Appendix A, we rederive these modified Maxwell's equations in the conventional 3D vector component notation, which will be more familiar to many readers of this section as compared to the relativistic Einstein notation that is typical in the field theory literature. Readers who are willing to accept the modified Maxwell's equations without derivation can proceed to Sec. 3. It should be noted that it more conventional treatments of magnetoelectrics [9, 10, 17] the magnetoelectric properties are introduced into the constitutive equations for the material and not into the Maxwell's equations directly. In this regard the Maxwell's equations are not really "modified", but this is an effective description which is largely equivalent. We use it here for historical reasons [15] and the fact that it allows a direct perspective on how surface properties are modified by the axion physics.

## 3   Quantized response from symmetry considerations

The topological field theory and resulting modified Maxwell's equations contain the essential axion angle parameter $\theta$ that characterizes the state of matter. It can take on different values in the TI or in the vacuum of free space. From the form of Eqs. 3 and 4, one can see that the additional physics described by the axion term only depends on derivatives of $\theta$, e.g. in the equilibrium case the physics only manifests at surfaces. For instance, the final term of Eq. 4 $[\frac{2\alpha}{c}\nabla(\frac{\theta}{2\pi})\times\mathbf{E}]$ gives a contribution that has the form of surface Hall effect the size of which depends on the net change in $\theta$ across the boundary.

Constraints on the permissible values of the axion angle $\theta$ follow from system symmetries. The Lagrangian defines the action $\mathcal{S} = \int dt dx^3 \mathcal{L}$ and since all physical bulk observables depend on $\exp(i\mathcal{S}/\hbar)$ they are invariant to changes to $\theta$ modulo $2\pi$ in an infinite bulk crystal. Therefore due to the transformation properties of $\mathbf{E}$ and $\mathbf{B}$, if *either* $\mathcal{T}$ or $\mathcal{P}$ are present, $\theta$ is constrained to be not only zero (as it is conventionally non-magnetoelectric materials), but can take on integer multiples of $\pi$ without changing any of the bulk physics [34, 35]. For instance, an inversion operation takes $\mathbf{E}$ to $-\mathbf{E}$ and hence $\theta$ to $-\theta$. As $\theta$ is defined modulo $2\pi$, an inversion symmetric system's $\theta$ must satisfy $\theta = -\theta + n2\pi$ and hence $\theta = n\pi$, where $n$ is an integer. Similar considerations hold for $\mathbf{B}$ and $\mathcal{T}$ symmetry. In fact, it can be shown that any magnetic point group that contains a proper rotation composed with $\mathcal{T}$, or an improper rotation without $\mathcal{T}$, constrains $\theta$ to be an integer times $\pi$ [36, 37]. Three-dimensional insulators with axion angles predicted to be an integer times $\pi$ can be further divided into two classes, which correspond to situations when $n$ is even (conventional) or odd (topological) [4]. As mentioned above, the difference between even and odd $n$ corresponds to the strong $Z_2$ index of TIs. With a change in $\Delta\theta$ across a surface from a TI to a conventional material, one gets a contribution to a surface Hall conductance that is

$$G_{xy} = \frac{\Delta\theta}{2\pi}\frac{e^2}{h} = \left(N + \frac{1}{2}\right)\frac{e^2}{h}, \tag{5}$$

where $n = 2N + 1$. As we will see below, $N$ indicates the number of fully filled Landau level (LL) or Chern layers on the surface when $\mathcal{T}$ is broken weakly[1].

---

[1]Note that nothing prevents surface Hall conductances of $N\frac{e^2}{h}$ from being on the surface of conventional insu-

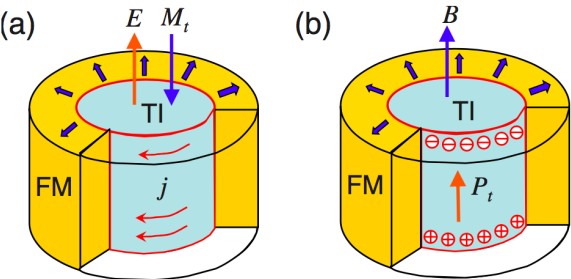

Figure 2: a) Magnetization can be induced in the same direction as the electric field for a TI in a cylindrical geometry. With the magnetization of a ferromagnetic layer pointing outward from the side surface of the TI a circulating current is induced by the electric field. This surface current is indistinguishable from a bulk magnetization. b) A charge polarization can be induced by a magnetic field directed along the cylinder axis. As magnetic field is turned on, an electric field is induced which drives charge to the end of the cylinder. Note that charge will be distributed over the whole end surface of the cylinder, not just the edge as displayed. From Ref. [4].

How a pure surface Hall conductance (e.g. $G_{xx} = 0$) then manifests as magnetoelectricity can be seen through a thought experiment. Consider a cylindrically shaped TI sample (Fig. 2a), which has an outwardly directed magnetic field large enough to induce a well defined Hall effect in the surface. Alternatively, one could imagine a magnetic layer deposited such that the magnetization is everywhere directed radially. With a pure Hall current, an applied electric field in the $\hat{z}$ direction, will induce a circumferential quantized Hall surface current $K_\phi$. As a surface current can be written as bulk magnetization e.g. $\mathbf{K} = \mathbf{M} \times \hat{r}$ and using Eq. 5, one has $M_z = \left(N + \frac{1}{2}\right)\frac{e^2}{h}E_z$ e.g. a magnetoelectric effect. Now consider the situation of a magnetic field that is turned on slowly from zero to a value $\mathbf{B}$ field in the $z$ direction. As the $\mathbf{B}$ field is being turned on, it induces a circumferential $\mathbf{E}$. With a pure quantized Hall response, a surface current will be driven in the $\hat{z}$ direction, where it flows to the ends of the cylinder giving a surface charge $\sigma_b$ as shown in Fig. 2b. A surface charge as such is equivalent to a bulk polarization via $\sigma_b = \mathbf{P} \cdot \hat{z}$. Integrating the current flow over the time scale that the magnetic field builds from zero to $\mathbf{B}$ gives $P_z = \left(N + \frac{1}{2}\right)\frac{e^2}{h}B_z$. Note that essential to maintaining an equilibrium polarization, is that after the magnetic field reaches its maximum and the circumferential electric field goes to zero, the surface charge cannot dissipate under its own field (in this idealized case) due to the lack of a longitudinal conductance e.g. the charge is "trapped" at the ends of the cylinder. This anticipates our below discussion in Sec. 6 on the important role of having only a dissipationless surface Hall current in order to define a true magnetoelectric e.g. diagonal conductance terms have to be vanishingly small. The fact that the response coefficient is the same for applied electric and magnetic fields is a well-known property of magnetoelectrics [6].

Although it is usually said that TIs are protected by $\mathcal{T}$, in fact – as discussed above – other symmetries can be equally important in quantizing $\theta$. However, since $\mathcal{P}$ and at least some rotation symmetries must be broken at any surface $\mathcal{T}$ symmetry is unique in protecting the existence of metallic surface states in TIs when it is present. Moreover, when the surface states are ungapped they prevent the observance of any magnetoelectric effects. Consider a

---

lators, but unless a system has its bulk and surface topological properties protected by crystalline symmetries (e.g. mirrors) as in the case of, for instance, topological crystalline insulators, such surface conducting layers will not be robust.

situation when $\mathcal{T}$ is broken only at the surface, by say a magnetized layer at the surface but inversion is preserved in the bulk. If the sample is thick enough, then the effect of $\mathcal{T}$ breaking at the surface can hardly affect the bulk of the material and whatever contribution the bulk has to be the same whether the magnetized layer is there or not. Therefore when $\mathcal{T}$ is *unbroken* at the surface and the surface is a metal, that surface is guaranteed to have a half-quantized surface anomalous Hall effect that exactly cancels the bulk quantized Hall effect. $\mathcal{T}$ must be broken in order to gap the surface and allow the bulk axion effect to manifest.

# 4 Analogy to (ferro)electric polarization

Although the treatment in Sec. 3 may seem straightforward, there are a number of aspects that should raise the eyebrows of experienced readers. First, we wrote that the Hall conductance of a surface can be $\left(N + \frac{1}{2}\right)\frac{e^2}{h}$. The $\frac{1}{2}$ is anomalous as we know from Thouless and collaborators [38] that the Hall response of a 2D gapped insulator *must* be an integer times $\frac{e^2}{h}$ as the Brillouin zone (BZ) integral of the Berry-curvature flux is quantized. Of course, in conventional insulators the integer is zero. How can it be half-integer here? Second, magnetoelectrics conventionally occur in materials that break both $\mathcal{T}$ and $\mathcal{P}$. But as we discussed above if either $\mathcal{T}$ or $\mathcal{P}$ is preserved then the magnetoelectric response will be quantized if inversion is preserved in bulk. So what does it mean to define a ME response in a material that has inversion? And how can it be that we can have half-quantized Hall response exhibited at the surface? It turns out these two aspects are related! Before delving into this too deeply, we make an illuminating and extended analogy to the related physical case of electric polarization.

## 4.1 Polarization in one dimension: the simplest topological scheme

In most textbook treatments of electric polarization, we are told that to compute polarization one must first identify a microscopic dipole and then average this dipole over space to obtain the macroscopic polarization vector $\mathbf{P}$. As $\mathbf{P}$ is defined as the electric dipole moment per unit volume, a natural definition is then

$$\mathbf{P} = \frac{1}{V_{cell}} \int_{cell} \mathbf{r}\rho(\mathbf{r})d\mathbf{r}, \tag{6}$$

where the integral is over the unit cell and $\rho$ is the microscopic charge density. The problem with this approach is that it depends on the definition of the unit cell. Indeed depending on the unit cell basis, completely opposite values of $\mathbf{P}$ may result. Another possible definition for polarization is where the volume integral and averaging volume in Eq. 6 are replaced by the sample volume itself. Although this is a straightforward procedure for a molecule whose density vanishes at infinity, such a definition is problematic for an infinite crystal. Moreover for a finite piece of a periodic crystal, the integral will have contributions from both the surface and the bulk, which gives the problematic situation that the quantity $\mathbf{P}$ which is supposed to represent a bulk macroscopic property of a crystal depends on surface terminations. These kind of ambiguities led to discussion for many years about whether or not electric polarization (and related quantities like pyroelectricity, piezoelectricity and the Born effective charge) could be defined as intrinsically bulk quantities or were determined by the surface termination [33, 39–41].

The difficulties with these conventional views can be highlighted by considering the case of a simple 1D chain of Na$^+$ and Cl$^-$ ions. Consider Fig. 3a (Type I lattice). The application of a 1D version of Eq. 6 would mandate that we choose a unit cell as for instance given by the box in Fig. 3a, which gives a dipole moment $\mathbf{d}$ and then average over a lattice vector

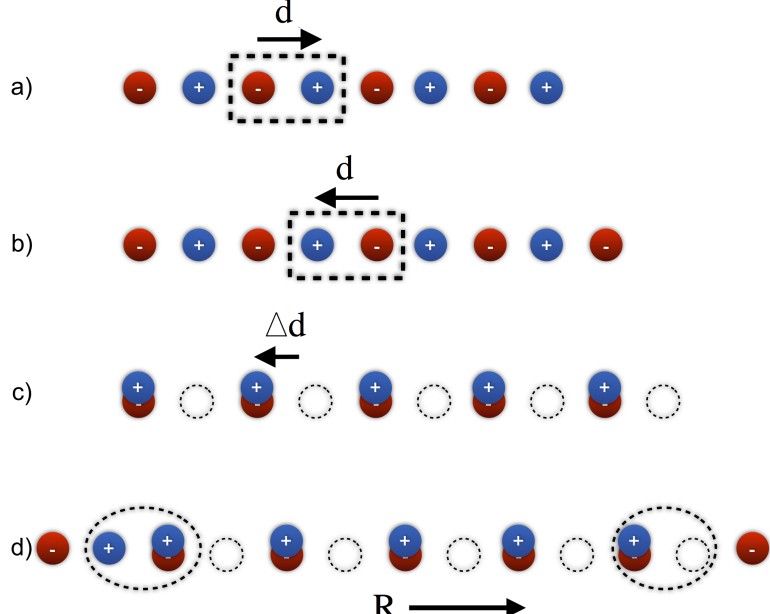

Figure 3: Polarization in a 1D inversion symmetric lattice. a) One choice of microscopic dipole **d** in the Type I lattice. b) Another choice of **d** that results in completely opposite polarization vector. c) A Type II lattice is the other possibility for an inversion symmetric lattice. The change in the microscopic dipole is $\Delta$**d**. Here the idea is that positive and negative charges are sharing the same lattice site. They are shown as slightly displaced for illustration purposes. d) Sandwiching a Type II lattice between sections of a Type I lattice gives charges of $\pm\frac{e}{2}$ at the interfaces, which one can see by allowing the point charges to be blurred out a bit in space.

**R.** For the choice of **d** shown, the polarization of the lattice is $\frac{e}{2}$. The problem is that the unit cell as defined in Fig. 3b is an equally valid choice, which gives the completely opposite value of the macroscopic polarization! The conclusion to be reached from this example is that it is impossible to use knowledge of a periodic charge distribution to give a unique value of polarization.

Similar difficulties in 3D were pointed out as early as 1974 [42], and were only resolved with what is now called "The Modern Theory of Polarization" [25, 26, 28, 30, 33, 41], in which it was realized the bulk polarization is a multivalued function that can only be defined modulo a polarization quantum $\mathbf{P}_q$. This lead to a new perspective that one should usually concern oneself with changes in polarization rather than with the polarization itself, as changes in **P** are well-defined and can be compared to experimentally measurable quantities.

We realize that for the 1D case shown in Fig. 3a and b, that despite the ambiguities and irrespective of which microscopic dipole is chosen, for a given 1D lattice only certain polarization values are possible and that for the Type I structure these values themselves form a 1D *lattice* whose nodes are $\frac{e}{2} \pm ne$, where $e$ is the electric charge and $n$ is an integer. The $\frac{e}{2}$ offset is an intrinsic property of this inversion symmetric lattice, which has additional significance that we come back to below. The multivaluedness is a natural consequence of the periodicity of a bulk crystal. It also suggests a definition for polarization that is in keeping with what is actually measured when a material undergoes a ferroelectric transition which is a *change* in polarization. An experimental determination of the spontaneous polarization is normally extracted from a measurement of the transient current flowing through the sample during a

switching process with a polarization change defined as

$$\Delta \mathbf{P} = \mathbf{P}(t) - \mathbf{P}(0) = \int_0^t dt \, \mathbf{J}(t). \tag{7}$$

Note that for the 1D crystal shown in Fig. 3a and b, zero is not a possible value of the polarization lattice. This may be surprising as by inspection, any ionic site in this structure is an inversion center and conventional wisdom says that in a centrosymmetric lattice one can not define a polarization. But the conventional wisdom is wrong! In the modern view, polarization can be defined, but it can take on only certain discrete values that are constrained by symmetry. The centrosymmetric constraint requires only that the polarization must get mapped onto itself by the inversion operation, which it can *only* do if polarization is a multivalued quantity. Here the inversion operation takes the 1D polarization $\frac{e}{2}$ to $-\frac{e}{2}$, which in the bulk is equivalent to $\frac{e}{2}$ via the 1D polarization quantum $e$. This is an extension (or caveat, if you will) to Neumann's principle, which usually states that, if a crystal is invariant with respect to certain symmetry operations, any of its physical properties must also be invariant with respect to the same symmetry operations [43,44]. Conventionally this would be taken to mean that a crystal with inversion symmetry must have $\mathbf{P} = 0$. However with the realization that polarization is a multivalued quantity, polarization in such systems can be non-zero because two values of the polarization that are separated by the polarization quantum represent the same bulk state. This multivalued polarization lattice is called the *formal polarization.*

The lattices shown in Fig. 3a and b are not the only centrosymmetric lattices using such ions, and one could also imagine a (fictional) lattice where we have moved the Na$^+$ and Cl$^-$ ions relative to each other by half a lattice constant such that they sit on top of each other as shown in Fig. 3c (Type II). This crystal structure gives a polarization lattice that is $0 \pm ne$. Note that whatever the difference between the Type I and Type II lattices are, it is not symmetry as their symmetries are identical. This gives a number of interesting consequences. First, if we imagine a structure of Type II sandwiched between two strings of ions in Type I lattice, we realize such a situation results in localized charges at the interface that are *quantized* as $\pm \frac{e}{2}$. In order to see this easily, one should imagine a lattice made of base units in which the charge is slightly spread out in space as shown in Fig. 4(top left). Then as in Fig. 3, one can construct two different kinds of lattices where these positive and negative charge units are either on top of each other or displaced by half a unit cell (Fig. 4(top right) and (middle left)). Sandwiching Type I between two copies of Type II gives localized charges at the interface that each have total charge $\pm \frac{e}{2}$ as seen in Fig. 4(middle right) and (bottom). Secondly, a general result follows in that any inversion symmetric structure has surface charges that are $n\frac{e}{2}$, with the two types of lattices giving two possibilities, where $n$ is a positive or negative integer. One can imagine for instance a symmetry transformation where you pass positive and negative charges through each other by moving them relative to each other by one lattice constant. This leaves the bulk invariant, but will increase the end charges by integer amounts. However, the surface charge will always quantized as $ne/2$ for a Type I lattice irrespective of the surface termination by extra positive or negative charge. For instance compare Fig. 3a or b where a has well defined polarization being charge neutral, but b does not. Both have exactly quantized $ne/2$ surface charges.

## 4.2 Formal polarization vs. effective polarization

It is important to note in all this discussion that one must distinguish between the formal polarization that we have been discussing and the actual effective polarization of finite sized crystallite with a particular surface termination. The former is a multivalued quantity, and the latter is a single valued quantity, which assumes one of the values allowed by the formal

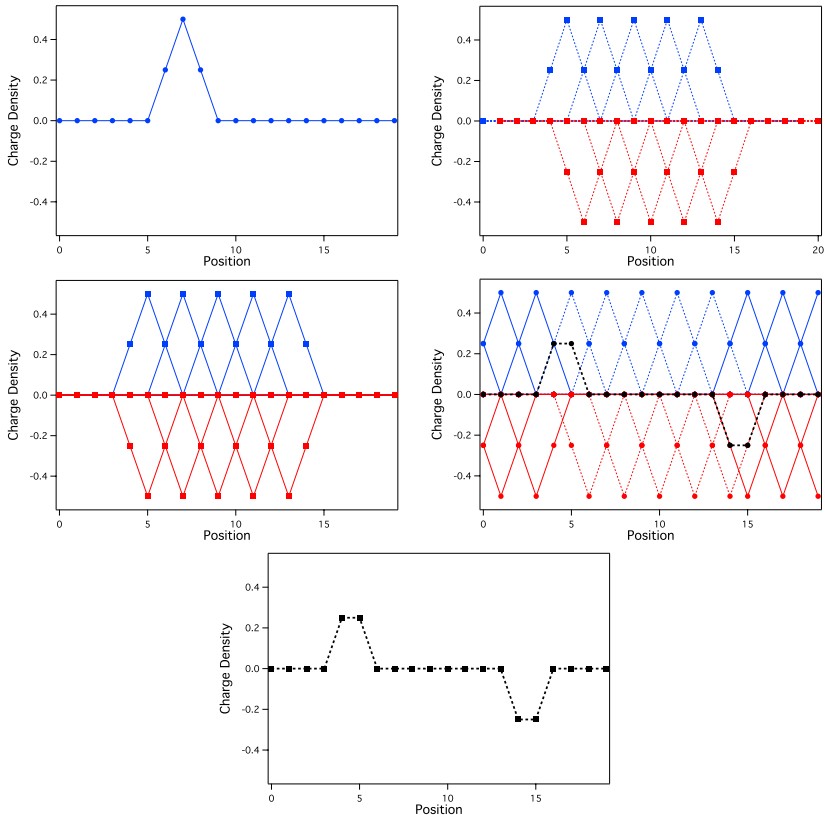

Figure 4: Simple 1D model to illustrate $\frac{e}{2}$ on the ends of a 1D chain. (top left) The simple "ion" unit that has +1 integrated unit of charge spread out over space. (top right) Charge unit arranged in Type I lattice with alternating positive and negative charge units. Note that system is neutral in bulk. (middle left) Charge units arranged in Type II lattice with positive and negative charge units on top of each other. (middle right) Type I lattice sandwiched between pieces of Type II lattice. Black represents the sum at each point of the net local charge. (bottom) Replotted net local charge. Integrated charge density at ends of chain shows that $\frac{e}{2}$ charges are located at the interface of the Type I and Type II lattices.

polarization. However, note that while the formal polarization is always well defined, for the effective polarization to be well defined the crystallite must have surfaces whose charge sums to zero. The polarization of a charged object depends on the choice of origin and a unique value for the effective polarization cannot be given[2]. In this regard the crystallite shown in Fig. 3b has a negative net charge and although an effective polarization cannot be defined, its formal polarization is still defined and it still has quantized end charges! In fact a measurement of the end charges is sufficient to determine the formal polarization even for a charged crystallite. Related to this, the effective polarization can only be finite if the crystallite breaks inversion. In this regard, the effective polarization of the crystallite shown in Fig. 3c is zero, which is one of the allowed valued of its formal polarization.

For the Type I lattice above, one may ask by what mechanism has the charge fractionalized? Where has the other half of the surface charge gone? Clearly, it has been swallowed in the

---

[2]This can be easily shown. An objects dipole moment can always be defined as $\mathbf{d} = \int \mathbf{r}\rho(\mathbf{r})d\mathbf{r}$. If one displaces the origin by an amount $\mathbf{r}_0$ then this quantity becomes $\mathbf{d}' = \int (\mathbf{r} - \mathbf{r}_0)\rho(\mathbf{r})d\mathbf{r} = \int \mathbf{r}\rho(\mathbf{r})d\mathbf{r} - \mathbf{r}_0 \int \rho(\mathbf{r})d\mathbf{r} = \mathbf{d} - \mathbf{r}_0 Q$, where $Q$ is the object's total charge. Hence, for finite $Q$ the dipole moment depends on the choice of origin.

SciPost Phys. **6**, 046 (2019)

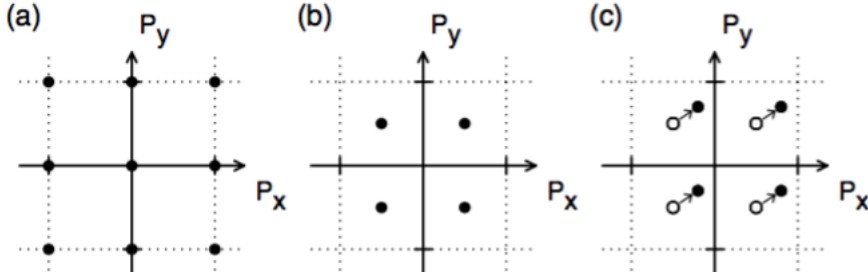

Figure 5: (a) and (b) The two possible 2D polarization lattices that are consistent with 2D square lattice symmetry. They correspond to the 2D projections of $BaTiO_3$ and $KNbO_3$ respectively. Note that for '(b)' zero is not a possible polarization. (c) A change in polarization induced by some symmetry-lowering change of the Hamiltonian. Adapted from Ref. [28].

bulk as a consequence of charge neutrality leaving only part of it on the surface. There are a number of similar models in condensed matter physics, where fractionalization happens through loosing part of an otherwise discrete unit into the bulk of the material. The Su-Schrieffer-Heeger (*SSH*) model applied to charge fractionalization in polyacetylene [45] and the spin-$\frac{1}{2}$ end spins that arises in a 1D spin-1 chain described by the Haldane model are prominent examples [46]. Similar effects have even been discussed in the context of physical chemistry, where it is known for centrosymmetric stereoregular oligomers (e.g. a molecular complex), that the end charges can only be integer multiples of 1/2 [47]. With both integer and half-integer terminations possible in 1D inversion symmetric lattices, the prescient reader may intuit that this discussion is remarkably similar to the notion that there are two families of inversion symmetric 3D insulators that are not distinguished by their symmetries, one of which has a *half*-quantized QHE on their surfaces. The prescient reader would be jumping ahead, but indeed this is exactly the point! Note that that the quantization of the end charges depends on symmetry. If one considers an inversion symmetry broken lattice where the + and - charges are at relative positions somewhere intermediate to Fig. 3b and c, then the end charges will be no longer quantized. The symmetry is essential to quantization. It will be the same in topological insulators.

### 4.3 Polarization in higher dimension

This example of a 1D ferroelectric can be easily extended to higher dimension. In keeping with its multivalued nature, the formal polarization can be expressed as

$$\tilde{\mathbf{P}} = \mathbf{P} + \frac{e\mathbf{R}}{V_{cell}}, \tag{8}$$

where $\mathbf{R}$ is a lattice vector $\mathbf{R} = \sum_j m_j \mathbf{R}_j$ and $\mathbf{P}$ is a value that depends on details of the crystal structure. However, similar to the 1D case, for inversion symmetric structures it is either zero or a value that corresponds to $\frac{e}{2}$ per surface unit cell. For a 2D inversion symmetric lattice with a square lattice symmetry there are two possible polarization lattices, which we represent in Fig. 5a and b. In a real world example, compare the cases of the ferroelectrics $BaTiO_3$ and $KNbO_3$ in their high-temperature cubic inversion symmetric structures. First principles calculations reveal that $\mathbf{P}$ for $BaTiO_3$ can be zero, whereas for $KNbO_3$ it can be $\frac{1}{2}\frac{e}{a^2}$ (where $a$ is the lattice constant) [28]. This is remarkable because the point group symmetries of both these lattices are exactly the same e.g. centrosymmetric, cubic etc. (Pm$\bar{3}$m), but they have different

symmetry protected constrained charge due to the different Wyckoff positions occupied. It is not possible to determine from symmetry alone which of the representations of the formal polarization is expressed in each lattice type[3]. Moreover, in each case, the formal polarization of a state is not just some value $\mathbf{P}$, but corresponds to a lattice of values that are related to each other by the polarization quantum $e\mathbf{R}/V_{cell}$. The surprising and important thing is that despite the fact that $KNbO_3$ is cubic and inversion symmetric, *none* of the allowed values of the formal polarization are zero! Similar to the 1D case, this is allowed because the inversion operation takes $\tilde{\mathbf{P}}$ to $-\tilde{\mathbf{P}}$, but these are related to each other by the polarization quantum (and indeed can even be said to be the same if we regard the allowed values of $\tilde{\mathbf{P}}$ as a multivalued quantity). See Resta and Vanderbilt [28] for further discussion on these points.

The realization that the formal polarization is a multivalued quantity and different crystal structures with the same symmetries can be intrinsically different again leads to the insight that experimentally, changes in the effective polarization can be defined with respect to a reference state. This change may be found across an interface to give a surface charge ($\sigma_b$) according to the expression $\sigma_b = \hat{\mathbf{n}} \cdot \left(\tilde{\mathbf{P}}_1 - \tilde{\mathbf{P}}_2\right)$ that given two actual materials gives a well defined value, with however the formal polarization being defined only modulo $\frac{e}{A_{cell}}$ where $A_{cell}$ is the surface unit cell area. Or it may be changes in polarization as a function of temperature, when undergoing a ferroelectric transition. In such a case the polarization lattice vectors uniformly shift as shown in Fig. 5c. Although given a new perspective in the context of topological properties by Niu [50], similar physics had been established since at least the 1960s regarding the physics of surfaces where it was known that for inversion symmetric crystals surface charge was always quantized in units of half-integer charge per surface unit cell [51–53].

## 4.4 Wannier functions and Berry's phase

These simple cartoon of point charges can be formalized with a mapping onto Wannier centers. Wannier functions are localized functions, which span the same Hilbert space as the extended Bloch states $|\psi_{n\mathbf{k}}\rangle$. They are defined as

$$|w_{n\mathbf{R}}\rangle = \frac{V_{cell}}{(2\pi)^3} \int d\mathbf{k} e^{i\mathbf{k}\cdot\mathbf{R}} |\psi_{n\mathbf{k}}\rangle, \tag{9}$$

where $\mathbf{R}$ are the lattice vectors. From the Wannier functions $|w_{n\mathbf{R}}\rangle$, one can define "Wannier centers" as $\mathbf{r}_{n\mathbf{R}} = \langle w_{n\mathbf{R}}|\mathbf{r}|w_{n\mathbf{R}}\rangle$. One can show that the Wannier centers can be written as $\mathbf{r}_{n\mathbf{R}} = \frac{V_{cell}}{e}\mathbf{P}_n + \mathbf{R}$, where $\mathbf{P}_n$ is the contribution to polarization of the $n$th band, which is the analog of Eq. 8 above [28]. Because the Wannier functions form a set of states that are only differentiated by the lattice vectors $\mathbf{R}$, the polarization inherits this indeterminacy by a quantized amount. Via the "Berry-phase theory of polarization" [25, 28, 33], the polarization $\mathbf{P}_n$ can be expressed as

$$\mathbf{P} = \sum \mathbf{P}_n = \frac{e}{(2\pi)^d}\text{Im}\sum_n \int_{BZ} d\mathbf{k}\langle u_{n\mathbf{k}}|\nabla_{\mathbf{k}}|u_{n\mathbf{k}}\rangle, \tag{10}$$

where the $|u_{n\mathbf{k}}\rangle$'s are the periodic part of the Bloch states of the $n$th occupied band and $d$ is the dimensionality. As the integrand of Eq. 10 is the Berry connection (a quantity whose integral

---

[3]The fact that these states with the same symmetries can have very different topological properties has an analog in topological electronic states, as a particular crystal space group is consistent with a number of distinct atomic Wyckoff positions and obviously lattices with different atomic positions can have very different properties. The role of the Wyckoff positions has been emphasized recently in systematized approaches to find new topological materials [48, 49].

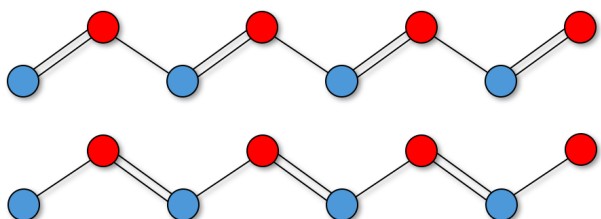

Figure 6: Two different patterns of dimerization in the $SSH$ model representing (top) $\delta t > 0$ and (bottom) $\delta t < 0$. Sublattices A and B are represented by blue and red respectively. The unit cell of A and B together represent a well defined unit that has a well defined connection to the molecular limit. Double lines indicates a double covalent bond making the atoms are closer and hence the hopping stronger.

is the Berry phase) Eq. 10 can be written for a single band as

$$\mathbf{P} = \sum \mathbf{P}_n = \frac{e}{(2\pi)^d} \text{Re} \int_{BZ} d\mathbf{k} \cdot \mathcal{A}, \tag{11}$$

where $\mathcal{A} = i\langle u_{n\mathbf{k}}|\nabla_{\mathbf{k}}|u_{n\mathbf{k}}\rangle$ is the Berry connection. As far as the polarization is concerned, the formation of Wannier functions can be regarded as an effective mapping of extended wave-functions onto a lattice of point charges that is in correspondence with the simple cartoon presented in Fig. 3 above[4]. The Berry's phase formulation of polarization makes explicit the polarization quantum as this just manifests through the phase's inherent $2\pi$ indeterminacy. One should also point out that the integrals in Eqs. 10 and 11 are gauge invariant (modulo the polarization quantum) despite the fact that their integrands are not gauge invariant, as they depends on the choice of the phases of the $|u_{n\mathbf{k}}\rangle$'s. In 3D, polarizations calculated in this fashion can be computed with modern ab initio packages.

In 1D the polarization can be calculated by integrating the Berry connection of the occupied states over the BZ [54] to get the Berry phase. Just like the axion angle that characterizes the ME coupling, the 1D polarization is perhaps most naturally expressed as an angle. It was noticed by Zak that when inversion symmetry is present this "Zak phase" becomes quantized and can only assume values of 0 or $\pi$ (modulo $2\pi$). The simplest 1D model of a topological insulator is that of the celebrated $SSH$ model [45]. One considers a Hamiltonian of spinless fermions hopping on a 1D lattice with staggered hopping amplitudes such as

$$\mathcal{H}_{SSH} = \sum_i (t + \delta t) c_{Ai}^\dagger c_{Bi} + (t - \delta t) c_{Ai+1}^\dagger c_{Bi} + \text{h.c.}. \tag{12}$$

The unit cell has a two atom basis labeled $A$ and $B$ with weak and strong hoppings. $\delta t$ controls the pattern of dimerization as shown in Fig. 6. The two phases are separated by a gap that is controlled by the sign of $\delta t$. The ground state Bloch function for this Hamiltonian is

$$|u_k\rangle = \frac{1}{\sqrt{2}} \begin{bmatrix} 1 \\ -e^{i\phi_k} \end{bmatrix}, \tag{13}$$

where $\phi_k$ is

$$\tan \phi_k = \frac{(t - \delta t)\sin k}{t + \delta t + (t - \delta t)\cos k}. \tag{14}$$

---

[4]This treatment uses the language of non-interacting Bloch functions, but one may also note that many-body formulations for the macroscopic polarization as a Berry phase have been given as well [41].

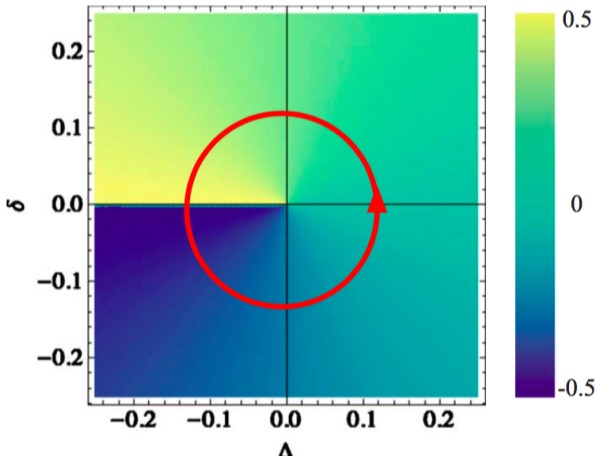

Figure 7: Polarization as a function of parameters $\Delta$ and $\delta$ in the 1D Rice-Mele model. Here the units are $ea$ where $a$ is the lattice constant. The line of discontinuity can be chosen anywhere depending on the particular phase choice of the eigenstate. From Ref. [27].

By evaluation of the Bloch wave's Berry's connection one finds that the Berry's phase integrated over the BZ is

$$\gamma = \oint dk \mathcal{A}(k) = \frac{\pi}{2}\left[1 + \text{sgn}\left(\frac{\delta t}{t}\right)\right] \tag{15}$$

and then by Eq. 11, $\mathbf{P} = 0$ for $\delta t > 0$ and $\mathbf{P} = e/2$ for $\delta t < 0$. The *SSH* model has a chiral symmetry that constrains these phases to have polarizations as such and fractionalized charge on the ends that sit at zero energy[5]. Polyacetylene itself does not have fractionally charged solitons because the molecular orbital states are occupied by two electrons nor does it have this chiral symmetry as it is broken by longer range hopping terms. It does however have inversion symmetry around the center of a bond which as we have discussed generally quantizes the polarization and the end charges of a 1D chain.

### 4.5 The 1D Thouless pump

One can break the inversion symmetry of the *SSH* model by introducing a term that breaks the onsite sublattice degeneracy. This Rice-Mele model [55] adds a term to Eq. 12 that has the form $\mathcal{H}_{RM} = \sum_i \Delta c_{Ai}^\dagger c_{Ai} - \Delta c_{Bi}^\dagger c_{Bi} + \text{h.c.}$ where $\Delta$ can be tuned from positive to negative. One can imagine starting from deep in the symmetry protected topological phase ($\delta t < 0$, $\Delta = 0$), but then sequentially changing both $\Delta$ and $\delta t$ such that inversion symmetry is first broken in the positive sense ($\Delta > 0$), then $\delta t$ is changed from negative to positive, then the sign of the inversion symmetry breaking term is flipped ($\Delta < 0$), then $\delta t$ is changed back to negative, and then finally inversion symmetry is restored with $\Delta = 0$ [27]. The Hamiltonian is returned back to its original configuration, yet if polarization is computed, one finds that it has changed by $\pm e$. Exactly one net elementary charge has been transferred through the system. Fig. 7 shows the computed polarization as a function of Hamiltonian parameters with the red circle

---

[5]For this 1D example of the Zak phase, one should point out that the Berry phase for each of the signs of $\delta t$ is a gauge dependent quantity e.g. it depends on the choice of the unit cell. However, given a choice of unit cell, the difference of Zak phases between the two states is uniquely defined and this determines the topological distinction between phases.

representing a smooth trajectory of these parameters. One elementary charge is transferred if a trajectory includes the central singular point. Closed trajectories that do not include this point do not transfer charge. The general mechanism is known as a Thouless pump [56] and its extension to higher dimension provides an explanation of quantized transport in topological systems. Although made generic when formulated in terms of Berry's phase the general notion was anticipated by Laughlin [57] in his gauge invariance argument for the quantum Hall effect (QHE).

Symmetry protected topological phases may also be considered from the perspective of adiabatic continuity. A symmetry protected phase can be said to be topological if it cannot be adiabatically deformed to the atomic limit while retaining its symmetries. In this regard, it is clear that only Type II in Fig. 3c above can be adiabatically connected to a well defined atomic limit (here actually the "molecular" limit of the fictitious symmetric $Na^+Cl^{-1}$ unit) in a fashion that preserves inversion. Therefore the Type I lattice is the topological phase. This idea that topological systems are ones that cannot be adiabatically connected to the atomic limit will be used again below.

# 5 The surface half integer Hall effect as a signature of a bulk magnetoelectric response

Aspects of the above discussion with regards to the lessons learned for polarization have direct analogy to topological insulators. Just as for case of the Type I and Type II centrosymmetric ionic chains, there are two kinds of insulators distinguished not by symmetry, but by topology. And in the same fashion that the Type I inversion symmetric 1D chain has half-quantized charges localized on its ends, a topological insulator with inversion symmetry (but broken $\mathcal{T}$) has a half-quantized QHE on its surface, whereas (in principle) a conventional insulator can only host a conventional integer QHE on its surface.

As suggested by these aspects and its definition, the formal magnetoelectric susceptibility can be formulated as a bulk quantity only modulo a quantum (here $\frac{e^2}{h}$) in much the same way as the formal electric polarization **P**. And in the same fashion, the formal magnetoelectric susceptibility is also properly expressed as a multivalued lattice. Note that its quantum $\frac{e^2}{h}$ is the same quantum as that found in the 2D QHE. As pointed out by Essin et al. [5], this follows from the fact that the smallest magnetic field that can be applied without destroying the periodicity of a crystalline system is one flux quantum per surface unit cell ($e/A_{cell}$). This can be combined with the flux quantum of polarization (one charge per surface unit cell) to give a natural quantum for the magnetoelectric susceptibility that is

$$\frac{\Delta \mathbf{P}}{\Delta \mathbf{B}} = \frac{e/A_{cell}}{h/(e/A_{cell})} = \frac{e^2}{h}. \tag{16}$$

## 5.1 A simple cartoon of a 3D topological insulator

As in the case for the polarization where **P** can be changed by a polarization quantum without changing anything of the bulk, the magnetoelectric susceptibility can be changed by $\frac{e^2}{h}$ while leaving the bulk invariant. Physically this corresponds to removing a quantum Hall layer from one surface (leaving behind a net quantum Hall layer of the opposite sign) and moving it through the system to the other side. The analogous operation in the 1D ionic chain is sliding the charges passed each other by one lattice constant, which changes both end charges but leaves the bulk invariant. Again by way of analogy with the 1D chain, this suggests a way of looking at inversion symmetric insulators as overlapping $\frac{e^2}{h}$ and $-\frac{e^2}{h}$ layers. As shown in Fig.

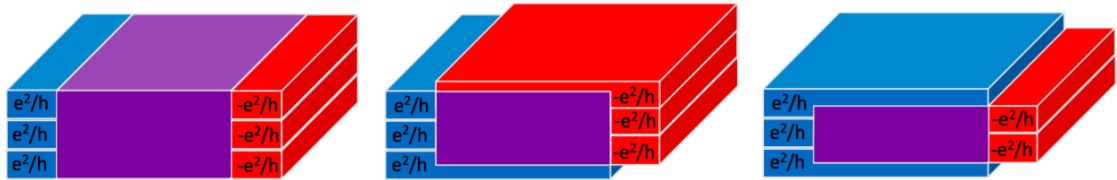

Figure 8: Cartoon model of conventional insulator and axion insulators built from overlapping Chern insulators. The ends of the diagrams are supposed to described some fictitious regions where the Chern layers are exposed. It is not supposed to represent a real system (although it could). (left) Chern insulators or quantum Hall layers of $\frac{e^2}{h}$ and $-\frac{e^2}{h}$ are centered on top of each other in a conventional insulator giving no net Hall response at any surface. Regions where Chern layers overlap are given in purple. (center) The distributions of layers in a topological insulator such that $-\frac{1}{2}\frac{e^2}{h}$ and $+\frac{1}{2}\frac{e^2}{h}$ are left on the surfaces giving the half-quantized Hall response of a surface and the quantized mangetoelectric effect. An applied electric (magnetic) field will give a magnetization (polarization) pointing in the same direction. Such a scenario would be realized if the surface magnetization was everywhere pointed outwards, or if a TI slab was placed in magnetic field, but the top and bottom surfaces were differentially doped to be electron and hole-like. (right) Here the topmost $-\frac{e^2}{h}$ layer has be removed such that $+\frac{1}{2}\frac{e^2}{h}$ is left over on the top surface. One can see that the bulk is not effected. A scenario as such is effectively realized in a TI slab in magnetic field.

8, one can conceive of conventional insulators as being materials these conducting layers are centered on top of each other and spatially overlap and cancel[6], whereas a TI is where layers of them are displaced from each other by half a unit cell, giving $\frac{1}{2}\frac{e^2}{h}$ on the surface. This picture gives immediate resolution to the issue raised above of how one can have a surface with a half quantized Hall effect, making clear the point that the surface of a TI is NOT a 2D system, but is the termination of a 3D material. This also answers the questions raised above about how one can get a half quantized Hall effect. The half quantized Hall effect is a bulk response expressed at the surface! The other $\frac{1}{2}\frac{e^2}{h}$ is lost into the bulk just as the $\frac{1}{2}$ surface charge we discussed above in the charge examples is lost in the bulk by virtue of bulk charge neutrality. One can imagine two scenarios of an inversion symmetric TI (Fig. 8(center) and (right)) where surfaces are terminated by Chern layers of the different or same signs of the Hall conductance. Different phenomena may manifest itself in either case, but the physics of the bulk is not changed. Similar to what we discussed above with the formal polarization vs. the effective polarization, we must distinguish between the formal magnetoelectric susceptibility and the effective magnetoelectric susceptibility. Only neutral objects can have an effective polarization defined, whereas the formal polarization is defined in any case. Similarly, an effective magnetoelectric susceptibility can only be defined when the net Hall response is zero. In this fashion the effective magnetoelectric susceptibility can only be defined for the crystallites in Figs. 8 (left) and (center). The crystallite in Fig. 8 (right) has a net Hall response and an effective magnetoelectric susceptibility cannot be defined, whereas its formal magnetoelectric susceptibility is independent of these considerations and can be defined.

This simple cartoon in Fig. 8 can be realized in a number of models. For instance, in the

---

[6]In 2D, canceling and spatial overlapping $\frac{e^2}{h}$ and $\frac{-e^2}{h}$ layers is precisely the situation in the topologically trivial 2D transitional metal dichalcogenides, where each $K$ and $K'$ valley host a Chern insulator with opposite quantized Hall conductance.

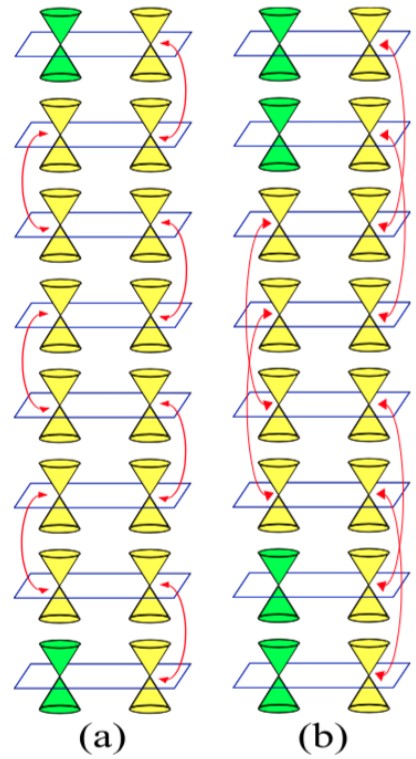

Figure 9: a) Schematic representation of the staggered interlayer mixing pattern of Dirac nodes. Interlayer couplings are represented by red arrows. The degenerate Dirac points at the ends of the red arrows mix and split, opening up a gap. A surface Dirac node colored green is left behind. b) A pair of surface Dirac nodes can arise as nodes are mixed in fashion separated by two layers. Mixing in this fashion requires a chiral symmetry.

context of creating a model for a chiral topological insulator Ref. [58] considered a model of bilayers of Dirac nodes coupled weakly both inter- and intra-bilayer. In the situation in Fig. 9a one considers a situation where a particular $R$ or $L$ Dirac node couples only to the layer above or below. This means that the $R$ Dirac points mix and split only within the bilayers, and the $L$ Dirac points, mix only between bilayers resulting in a staggered mixing pattern as shown. In this fashion the bulk becomes gapped but a single Dirac node is left on the surface. Fig. 9b represents a more complicated model for a *chiral* topological insulator where couplings between farther separated layers ultimately generated a greater number of Dirac cones on the surface. Pershoguba and Yakovenko [59] considered a 3D model related to *SSH* (called the Shockley model therein) where 2D A and B layers were coupled in a fashion similar to the ionic couplings in *SSH*. Mong et al. [60] explicitly considered model for an *antiferromagnetic* topological insulator that was effectively alternating magnetized Chern layers. Depending on the magnetization of the surface termination, the surface Hall conductance can be $\pm\frac{e^2}{h}$.

Considering all the above discussion, one can define the formal magnetoelectric susceptibility then as

$$\tilde{\alpha}_{ij} = \alpha_{ij} + \left(\frac{1}{2} + N\right)\frac{e^2}{h}\delta_{ij}. \tag{17}$$

Here $\alpha_{ij}$ is the magnetoelectric susceptibility from other contributions including spin and frozen-ion and lattice-mediated contributions and $N$ is an integer that corresponds to the num-

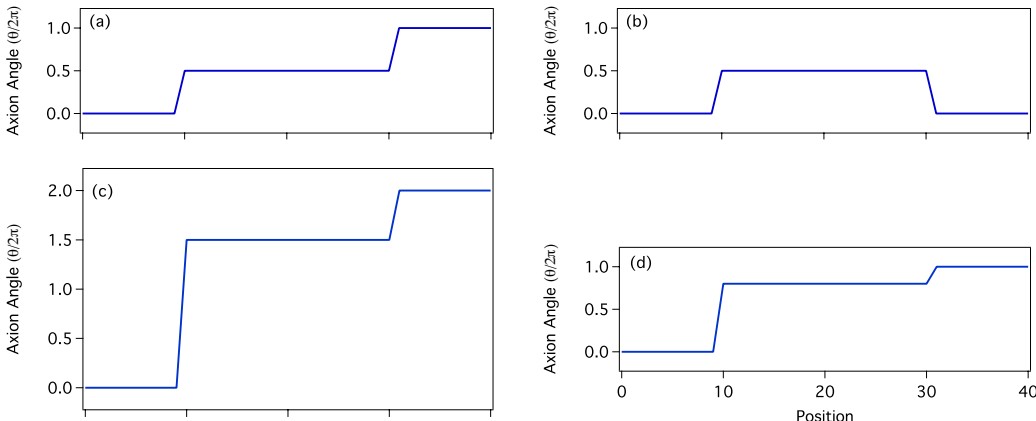

Figure 10: The axion angle $\theta$ as a function of space passing through a TI slab for four different scenarios. Jumps in the axion angle are shown as occurring at the sample surface positions. a) A TI slab in magnetic field, showing the minimal jumps of $\pi$ on each surface that corresponds to two Hall effects of $\frac{1}{2}\frac{e^2}{h}$ each. b) A TI slab with an inwardly directed magnetization for each surface c) A TI slab with a quantum Hall layer on the front surface in addition to the $\frac{1}{2}\frac{e^2}{h}$ Hall effect. d) An example of a TI slab for a material that does not have inversion symmetry. The jumps in the axion angle on the front and back surfaces are *not* quantized, giving surface fractional quantum Hall effects of arbitrary magnitude. However, the net winding of the axion angle is quantized in units of $2\pi$ and hence the total Hall conductance of the full slab is an integer times $e^2/h$.

ber of quantum Hall layers on the surface (or equivalently the number of filled Landau levels). In terms of the axion angle (defined modulo $2\pi$) one may write

$$\tilde{\alpha}_{ij} = \alpha_{ij} + \theta \frac{e^2}{2\pi h}\delta_{ij}. \tag{18}$$

The fact that the formal ME susceptibility is a multivalued function resolves the issue as to how a magnetoelectric can be defined in the presence of bulk inversion symmetry. Centrosymmetry requires that the response tensor can get mapped onto itself by the inversion operation, which if it is non-zero it can only do if the magnetoelectric susceptibility is a multivalued quantized quantity. Otherwise inversion symmetry would mandate that the magnetoelectric tensor is zero. But if it is multivalued, then the inversion operation takes $\frac{1}{2}\frac{e^2}{h} \rightarrow -\frac{1}{2}\frac{e^2}{h}$, but by the magnetoelectric susceptibility quantum this is equivalent to the original $\frac{1}{2}\frac{e^2}{h}$. Alternatively, one can say the formal ME susceptibility is set by $\theta$, which in the TI is $\pi$ (or odd integer multiples thereof). Due to the $2\pi$ periodicity of $\theta$, the ME susceptibility does not have to be zero if inversion is maintained in the bulk, but it is constrained to quantized values.

To get extra insight into the physical significance of the axion angle $\theta$ consider a hypothetical situation where all of space is divided by a slab of a TI. We know that the vacuum on either side has its $\theta$ value constrained to be $\pi$ times an even integer and if the TI has inversion symmetry then $\theta$ inside the TI is constrained to be an odd integer times $\pi$. This determines that the Hall conductance of each surface by itself is an odd integer times $\frac{1}{2}\frac{e^2}{h}$, but the total Hall conductance of the slab must be an integer times $\frac{e^2}{h}$. In Fig. 10 we show a number of different situations corresponding to various ways of breaking $\mathcal{T}$ symmetry on the surfaces of the TI or by their differing surface environments. In Fig. 10a, we show the situation that corresponds

to experiments performed so far where a TI slab is placed in magnetic field and exhibits a quantized Faraday rotation. Both surfaces show a quantum Hall effect of $+\frac{1}{2}\frac{e^2}{h}$. Fig. 10b corresponds closely to the situation envisioned originally in Ref. [4], in which a magnetized layer coats the TI in a fashion such that the magnetization points inward from both surfaces. In Fig. 10c, one envisions that due to finite doping the front surface has an additional $\frac{e^2}{h}$ quantum Hall layer Chern layer stitched to it.

These pictures gives additional insight into the relation between the conventional QHE and the topological magnetoelectric effect. As discussed above it is necessary to break $\mathcal{T}$ symmetry, but for instance preserve $\mathcal{P}$ to get a half quantized Hall effect on a surface. But the conventional Hall effect in a 2D electron gas effect does not require any such additional symmetries. One can see that in the limit where the TI slab thickness goes to zero, since the total change in $\theta$ must be an even integer times $2\pi$ (since there is the inversion symmetric vacuum on both sides), the total Hall conductance of a 2D layer must be an integer times $\frac{e^2}{h}$ irrespective of whatever happens in the bulk. Thus the axion electrodynamic formulation naturally turns into the conventional QHE in the limit that the slab becomes 2D. This discussion would also be relevant Faraday rotation experiments on films of a non-centrosymmetric TIs, where no improper rotation quantizes the bulk $\theta$ (HgTe has an $S_4$ symmetry (improper rotation) that presumably quantizes its bulk $\theta$ in recent experiments [61]). If inversion symmetry or an improper rotation does not quantize $\theta$, nothing mandates that its bulk axion angle is an odd integer times $\pi$ and hence although the *total* Hall conductance of the *entire* slab must be an integer times $\frac{e^2}{h}$, the Hall effect at either surface could be anything. See Fig. 10d for an example of how this might happen. An analogous version of Fig. 8b would be one where the positive and negative QH layers would be shifted by some amount that is not half a lattice constant allowing a non quantized Hall-effect on the surface, but still quantizing the sum of top and bottom Hall responses. This is only allowed if inversion is not a symmetry in the bulk of the material. If all of $\mathcal{T}$, $\mathcal{P}$, proper rotations composed with time-reversal, and improper rotation symmetries are broken and bands are still inverted, the ME susceptibility is likely to still be large through the same Chern-Simons mechanism, however it will not be quantized and an experiment that relies only on the sum of the Hall conductances from the two surfaces in magnetic field is not evidence for the quantized magnetoelectric effect.

This discussion hopefully makes clear that for materials that do have inversion or other relevant symmetries there is no fundamental difference between measuring a material that has magnetization directed in the same direction on both surfaces or inward/outward on both surfaces (Figs. 10a and b respectively). The latter has been explicitly called the axion state and said to be the configuration to measure the topological ME effect. [24, 62, 63]. Although the development of systems that realize this configuration is very important from a materials perspective, we do not believe it warrants any particular consideration as anything special or fundamental. Both scenarios have the same formal ME susceptibility. As shown in Fig. 10, the two configurations should just be considered as different experimental conditions and realize fundamentally the same thing. Both cases arise through the same $\mathbf{E} \cdot \mathbf{B}$ physics and as such they are simply different (partial) manifestations of the same physics. For instance, one can get the same dependence of $\theta$ on position as in Fig. 10b by instead of having an inward pointing magnetization on both surfaces, but instead putting the slab in a magnetic field, but then absorbing locally a Chern insulator layer with Hall conductance $-\frac{e^2}{h}$ layer on the back surface.

## 5.2 The formal magnetoelectric susceptibility vs. the effective magnetoelectric susceptibility

As we have discussed above, it is not important to break inversion to discuss the formal ME susceptibility in a bulk material. In fact, inversion quantizes the $\theta$ angle. However, in order to generate a macroscopic moment of a finite size sample, global inversion symmetry of the crystallite *must* be broken. For instance, because inversion symmetric $Bi_2Se_3$ in magnetic field breaks only $\mathcal{T}$, a slab of such material cannot exhibit a net macroscopic moment from magnetoelectricity unless inversion is broken macroscopically through some other means to get a finite magnetoelectric susceptibility. Moreover, as we mentioned – in an analogous fashion to the polarization discussion above – we can distinguish between the formal ME susceptibility and the effective ME susceptibility of an actual crystallite. Recall that in Fig. 3a and b both represent the same kind of inversion symmetric crystal (with the same formal polarization), but a macroscopic dipole moment can only be defined for Fig. 3a, as the crystallite represented by b is macroscopically inversion symmetric and charged. The effective polarization can only be defined for a charge neutral object. In exactly the same fashion, a topological insulator slab's effective ME susceptibility can only be finite if the total Hall conductance is zero. Also note that just like in the case of the 1D lattice where it is sufficient to measure just a single end charge to establish the formal polarization, it is in principle sufficient to measure the surface Hall effect of a single surface to establish the formal ME susceptibility. One can in principle do this by placing a TI in magnetic field and measuring the Faraday or Kerr rotation.

To break inversion and ensure a zero total Hall conductance, surfaces could be doped with charge species that make them differently electron and hole doped top or bottom, or coated by a magnetic layer that has magnetizaton outwards or inwards on both surfaces. It is important to note that as long as the inversion breaking field is local (whether this is a magnetic layer or preferential electron and hole doping on top and bottom) then $\theta$ should still be quantized in the bulk. In this regard, putting a sample in an $\mathbf{E}$ field to preferentially bias the surfaces with different signs will formally destroy the quantization although if the field is not too strong this inversion symmetry breaking effect will be likely weak. In the case relevant for our experiment [64], inversion symmetry constrains the crystal's bulk $\theta$ term to be $2\pi\left(N + \frac{1}{2}\right)$ but a net macroscopic moment cannot be generated, because the applied magnetic field does not break inversion, however, the sample can still be considered magnetoelectric in the sense that we have discussed above. To get a "true" magnetoelectric (e.g. the possibility to create a moment from an applied field) with a finite effective ME susceptibility one must have a situation like in Fig. 10b, that one would get by depositing a magnetic layer on both surfaces such that everywhere the magnetization of both surfaces points in or out (or by surface doping) [24, 63]. Then the sample is described by a particular $\theta$, the sample macroscopically breaks inversion such that it can have a ME effect, but the bulk is unaffected such that $\theta$ is quantized.

Systems that have a net winding of the $\theta$ angle across the bulk as shown in Fig. 10a or c will show a quantized Faraday effect, but no true magnetoelectric effect as the pattern of $\theta$ is consistent with inversion symmetry being maintained. Systems that have a dependence of the $\theta$ angle in the bulk as shown in Fig. 10b will show no Faraday effect, but will demonstrate true magnetoelectric effect (putting aside the finite $\omega$ effects discussed below) as the spatial dependence of $\theta$ demonstrates that inversion is broken. The effective ME susceptibility of such a system will be finite. And not to belabor the point, but we wish to emphasize that both scenarios in Fig. 10a or b are indicative of axion electrodynamics and the topological ME effect and simply different (partial) manifestations of the same quantized $\mathbf{E} \cdot \mathbf{B}$ physics. One is not more fundamental than the other as they have the same formal ME susceptibility.

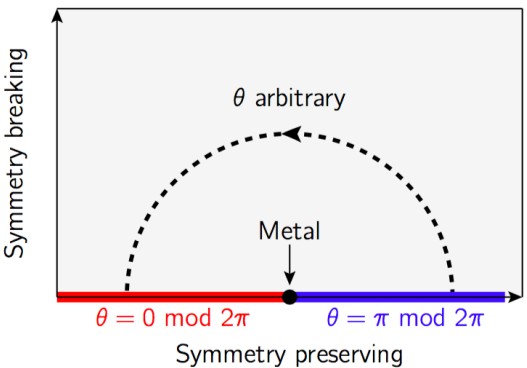

Figure 11: A schematic view of the possible values of $\theta$ as a function of both a symmetry preserving parameter and a symmetry breaking parameter in a Hamiltonian as considered in Ref. [29]. Along the horizontal axis $\theta$ jumps from $\pi$ mod $2\pi$ to zero mod $2\pi$. The gap must close along the horizonal axis, whereas the topological class also changes, but the gap remains finite if the path is on the dotted circular line.

### 5.3 The Thouless pump in 3D topological insulators and hybrid Wannier functions

Let us make a further connection between the 1D ionic chain and TIs. Independent of the particular realization, there are two ways to change the topological class of a symmetry protected topological phase. One can for instance, close a gap while preserving the symmetry. This is the situation considered for a number of topological phase transitions in topological insulators [65,66] by changing the energetic ordering of bands through a band inversion transition. The other possibility is to maintain the gap while changing a parameter that breaks the protecting symmetry and moves the system from a symmetric case through a symmetry broken regime and then back to a symmetric phase. The thought experiment we have considered above for the 1D lattice where positive and negative charges where moved past each other is an example of that. Starting from the centrosymmetric phase in Figs. 3a and b (Type I) and moving the relative positions of the charge to put it in the phase demonstrated in Fig. 3c (Type II) the system goes through a regime where the ions are at intermediate positions and breaks inversion. As mentioned above this phase with intermediate positions of the ions will not have quantized end charges, but remains insulating and keeps the gap throughout the transition. One unit net charge is pumped through the system in the process. Thouless charge pumping in the Rice-Mele model discussed above is another example of this.

For a 3D TI, analogous to the thought experiment of 1D charge pumping in the Rice-Mele model, Essin et al. [5] considered the model of Fu, Kane, and Mele [1] for a 3D topological insulator on the diamond lattice. They add a staggered Zeeman field on the two fcc sublattices, the magnitude of which can be written $|\mathbf{h}| = m\sin\beta$. For non-zero values of $\beta$ this term breaks both inversion and time-reversal, but for $\beta = 0$ and $\pi$ obviously breaks neither. If the nearest-neighbor hopping amplitude is set to be $3t + m\cos\beta$, one can smoothly vary the single parameter $\beta$ from zero to $\pi$ and drive the system between trivial and TI phases while keeping the gap constant. In so doing, Essin et al. find that the magnetoelectric polarizability interpolates smoothly between 0 and $e^2/2h$. The two end members are the two symmetry protected phases and both possess quantized ME responses as expected (quantized at zero in the trivial phase). In the intermediate regime, symmetries are broken and although the ME response is large, it is not quantized. Coh et al. [29] used the same general idea in their work looking for materials with large magnetoelectric couplings. They considered a model

Hamiltonian which depends on two parameters as shown in Fig. 11, one of which preserves time and/or inversion symmetry, and another that breaks these symmetries such that $\theta$ can assume a non-quantized value. Along the horizontal axis where a symmetry is preserved $\theta$ must jump discontinuously from $\pi$ mod $2\pi$ to zero mod $2\pi$. The gap closes at a point indicated "Metal". Another route is possible however and one could imagine taking a trajectory indicated by the black dashed line such that $\theta$ can vary smoothly and continuously without closing the gap anywhere along the path.

Similar to the 1D case given for the Rice-Mele model in Fig. 7 where a net loop in Hamiltonian parameter space "Thouless pumps" charge across a system, one can ask what happens when the $\mathcal{T}$ breaking parameter $\beta$ in the Essin extension to the Fu-Kane-Mele model is varied over a whole cycle from 0 to $2\pi$ [1,5]? In analogy with Fig. 7 we may expect that if a origin encircling loop is made in the space shown in Fig. 11, something is pumped, but what? To answer this question we need a little more formalism.

Just as the cartoon we had of the ionic chain could be formalized in terms of a Berry phase and made explicit in the form of the *SSH* model, the cartoon of QH layers shown in Fig. 8 can be made explicit. Vanderbilt and collaborators [67–69] considered a hybrid Wannier function representation obtained by Wannier transforming the Bloch functions in one dimension while keeping them extended and Bloch-like in the other two. The hybrid Wannier functions can be expressed as

$$|W_{nl_z}(k_x, k_y)\rangle = \frac{1}{2\pi} \int dk_z e^{i\mathbf{k}\cdot(\mathbf{r}-l_z c\hat{z})}|u_{n,\mathbf{k}}\rangle. \tag{19}$$

where $l_z$ is a layer index and $c$ is the lattice constant along $\hat{z}$. From this procedure, one can obtain the Wannier centers $\bar{z}$ and plot them as a function of the orthogonal momentum in the projected BZs. Under conditions that that these Wannier "sheets" do not touch, as each sheet corresponds to a filled 2D band, each sheet's $\hat{z}$ Berry flux is quantized to $2\pi$ times an integer by the Chern theorem [67]. As one varies the orthogonal momentum across the BZ, Wannier centers $\bar{z}$ can either return to their original values of $\bar{z}$ or they can be shifted by one lattice constant.

In the case that the $\bar{z}$'s cross the BZ, such a plot allows one to see how electrons are adiabatically pumped along $\hat{z}$ as $k_x$ and $k_y$ are varied. 2D quantum Hall systems, 2D quantum spin Hall insulators, and weak and strong 3D TIs can be characterized by examining how the Wannier center sheets connect along time-reversal invariant lines in the BZ for different "Wannierization" directions. How this occurs in a 2D system can be seen easily for the example of a single layer 2D Chern insulator in Fig. 12 (far left). As a function of $k_y$ the Wannier center moves in $\hat{z}$ and connects one unit cell to another. A 2D quantum spin Hall layer is effectively two copies of the same as shown in Fig. 12 (middle left). In the absence of spin-mixing terms, it shows the "switching of partners" characteristic of time-reversal invariant phases. A trivial 2D insulator would show bands that do not cross the unit cell in the $\hat{z}$ direction as a function of $k_y$.

In 3D, the behavior of the hybrid Wannier functions depends on the topological class. Because an isolated "Wannier sheet" represents a discrete 2D system with all occupied or unoccupied states, each holds a quantized amounts of Berry-curvature flux e.g. an integral Chern number. For weak TIs, the system looks trivial in at least one Wannierized direction. The non-trivial behavior of the hybrid Wannier function sheets in strong TIs will be apparent irrespective of the direction chosen to Wannierize. Hybrid Wannier function calculations for the 3D Fu-Kane-Mele model (a four-band model of $s$ orbitals on the diamond lattice with spin-orbit interaction) are shown in Fig. 12 (right). Here the $\hat{z}$ direction is taken to be [111]. The hybrid Wannier function representation makes explicit the fact that one cannot create Wannier functions in such topological systems that respect all symmetries, despite the fact that

Figure 12: (far left) Flow of Wannier charge centers along $\hat{z}$ vs. $k_y$ for a 2D Chern insulator. (middle left) Flow of Wannier charge centers along along $\hat{z}$ vs. $k_y$ for a 2D quantum spin Hall layer. (right) Flow of Wannier charge centers in hybrid Wannier representation for 3D 3D Fu-Kane-Mele model. From Ref. [67].

the eigenstates of Hamiltonian have the Bloch form [70,71]. And as was anticipated in our discussion of the 1D ionic chain and of Fig. 8 above, one cannot adiabatically connect the system to the atomic limit while preserving all symmetries. This is related to the fact that in a 3D strong topological insulators one goes from an 2D trivial insulator to a 2D topological insulator (or vice versa) in going from $k_z = 0$ to $k_z = \pm\pi$ as shown in Fig. 12 (right).

The hybrid Wannier function representation also makes quite apparent the Thouless pump mechanism for quantized transport in the 3D TIs. In general in the Thouless pump, one imagines replacing one of the momentum e.g. $k_y$, by some parameter $Q$ that characterizes a reduced dimensional Hamiltonian. For instance for Fig. 12(far left), as a charge's momentum $k_y$ is cycled from 0 to $2\pi/b$ the Wannier center is displaced by one or more unit cells in the $z$ direction. Thereby the quantized Hall transport of the 2D quantum Hall effect is mapped into a quantized 1D polarization induced by a cycle in $Q$ of the 1D Hamiltonian's parameters.

A similar situation applies for the 3D TI , where now the adiabatic pump corresponds to a pumping of a quantized amount of Berry flux across a unit cell. Using the staggered field that [5] added to the model of Fu, Kane, and Mele [1] for a 3D topological insulator on the diamond lattice (discussed above), Taherinejad et al. [68] showed that as $\beta$ is cycled from 0 to $\pi$ and then to $2\pi$, the system's bulk goes from trivial to topological and back to trivial, but in so doing the axion angle was pumped by $2\pi$. This is completely analogous to the charge pumping in the Rice-Mele model. Physically it corresponds to displacing Chern layers with quantized Hall conductance by a unit cell. This leaves a deficit of Chern conductance on one surface and an excess on the other. If one ran the $\beta$ pump from $\pi$ to $3\pi$ this would be equivalent to going from a situation which is Fig. 10b to a situation where the central region increases its $\theta$ from $\pi$ to $3\pi$. From the perspective of the hybrid Wannier functions, as $\beta$ is varied the pumping of $\theta$ by $2\pi$ occurs by a series of band touching events between Wannier sheets, such that one Chern number of Berry curvature flux is passed off to the neighboring sheet with each touching.

The magnetoelectric susceptibility coupling can be expressed in the spirit of the 1D polarization discussed above in terms of the Berry curvatures. In terms of the axion angle defined via Eq. 18, the susceptibility can be written as an integral over the Brillouin zone of the "Chern-Simons 3-form" as

$$\theta = -\frac{1}{4\pi} \int d^3k \, \epsilon^{ijk} \, \mathrm{Tr}\left[ \mathcal{A}_i \partial_j \mathcal{A}_k - i\frac{2}{3} \mathcal{A}_i \mathcal{A}_j \mathcal{A}_k \right]. \tag{20}$$

Here $\mathcal{A}_i$ is again the Berry connection in the $i$th direction and the trace is over occupied states. An arbitrary gauge transformation can be shown to only shift the 3-form integral by an integer times $2\pi$ [4,5,68] so again $\theta$ is best regarded as a phase angle that is only well-defined modulo $2\pi$. Thus again the presence of either time reversal or inversion requires that $\theta$ be an integer times $\pi$.

Table 1: Analogies between polarization and magnetoelectric susceptibility in inversion symmetric systems. For both cases presented here for inversion symmetric systems, $N$ is an even integer in trivial materials and is an odd integer in topological systems. Based in part on Ref. [72]. Note that both Chern forms can be expressed as angles.

| Quantity | 1D Polarization | 3D Magnetoelectric Susceptibility |
|---|---|---|
| Observable | $\mathbf{P} = \partial \langle H \rangle / \partial E$ | $\alpha_{ij} = \delta_{ij} \partial \langle H \rangle / \partial E_i \partial B_j$ |
| Surface "charge" | $N \frac{e}{2}$ | $N \frac{1}{2} \frac{e^2}{\hbar}$ |
| "Polarization" quantum | $\lvert e \rvert$ | $\lvert \frac{e^2}{\hbar} \rvert$ |
| Thouless pumped quantity | Integer charge | Quantized Berry flux (e.g. Chern layer) |
| Condition for effective $\mathbf{P}$ or $\alpha_{ij}$ | Charge neutral | Zero net Hall conductance |
| Chern form | $\gamma = \oint dk \mathcal{A}(k)$ | $\theta = -\frac{1}{4\pi} \int d^3k \, \epsilon^{ijk} \, \mathrm{Tr}\left[ \mathcal{A}_i \partial_j \mathcal{A}_k - i\frac{2}{3} \mathcal{A}_i \mathcal{A}_j \mathcal{A}_k \right]$ |

In this work we have explored in-depth analogies between polarization (in particular in 1D) and the 3D magnetoelectric susceptibility. We conclude this section with the summarizing Table 1 where we make the correspondences explicit between the various quantities.

# 6 The effects of residual surface dissipation on the magnetoelectric response of topological insulators

In the search for a "true" magnetoelectric with finite effective ME susceptibility effect in topological insulators, one may wish to measure the corresponding dc response e.g. a true macroscopic electric polarization when placed in dc magnetic field or a macroscopic magnetization when placed in a dc electric field. As we have seen above, in order to create a macroscopic moment symmetry considerations must apply globally to the sample i.e. the crystallite itself must break $\mathcal{T}$ and $\mathcal{P}$ independent of the local symmetries of the bulk. However, there are also dynamical considerations and it turns out that the requirements to see a dc effect puts severe limits on any residual dissipation in the surface states that occurs through imperfect gapping.

To analyze the effects of residual dissipation it is useful to use the language of a surface Hall effect albeit one that can exhibit a half quantized Hall conductance. As we have discussed above, in the dissipationless limit (e.g. small $G_{xx}$) this is equivalent to a bulk magnetoelectric effect. To be explicit consider again the geometry of a cylinder shaped TI, where a $\mathcal{T}$ breaking perturbation is applied such that $\mathcal{T}$ is broken at the surface as in Fig. 2. In the situation of a perfectly formed QHE on the surface of the TI where $G_{z\phi} = \left(N + \frac{1}{2}\right)\frac{e^2}{\hbar}$ and $G_{zz} = 0$, if an electric field is applied in the $\hat{z}$ direction, this will cause a surface current to flow in the circumferential $\phi$ direction. Again, an object with a surface current as such, is indistinguishable from a bulk magnetization $\mathbf{K} = \mathbf{M} \times \hat{n}$. As $\mathbf{K} = G\mathbf{E}$, one may write that $M_z = \left(N + \frac{1}{2}\right)\frac{e^2}{\hbar}E_z$. As discussed above, there is a reciprocity for magnetoelectrics and the same response function (in the low frequency limit) is relevant with applied magnetic field. As discussed above, in this case, as a **B** field is turned on, it induces a circumferential electric field that due to the Hall effect drives a current in the $\hat{z}$ direction. This charge is "trapped" at the ends of the cylinder due to lack of longitudinal conductance. Hence a polarization forms in response to an applied magnetic field.

But what happens in the presence of finite $G_{zz}$, which can occur due to insufficient localization of the surface states? Qualitatively, in the presence of an applied $\hat{z}$ axis electric field a finite $G_{zz}$ will allow some current in the $\hat{z}$ direction giving a surface charge at the cylinder ends that will eventually cancel the applied electric field. Similarly for an applied $\hat{z}$ axis magnetic field, a finite $G_{zz}$ will allow charge to leak out of the ends of the cylinder allowing the effective polarization to dissipate after the magnetic field is ramped to its final value. We realize from this qualitative discussion, that the capacity to build up magnetization or polarization in the presence of finite $G_{zz}$, is a matter of time scales, and the effect of finite non-zero $G_{zz}$ may be ameliorated at high enough frequencies.

Starting from a scenario of **B** applied at a low frequency $\omega$ along $\hat{z}$ and realizing that a finite $G_{zz}$ gives a channel whereby built up $P_z$ polarization can dissipate, one can show for the cylinder geometry that

$$P_z = \frac{G_{z\phi}}{1 - \frac{i2G_{zz}}{\epsilon_0 R \omega}} B_z, \tag{21}$$

where $R$ is the radius of the cylinder. Since $G_{zz}$ can never be identically zero, it is unreasonable to imagine that the true $\omega = 0$ dc magnetoelectric susceptibility is finite. For a $R \sim 1$ mm cylinder, $G_{zz}$ has to be on the order of $10^{-7}\frac{e^2}{h}$ to push the frequency crossover down well below the kHz range of conventional ac susceptometers. Note that in addition to any surface dissipation, any residual bulk conduction (commonly present in TIs) also adds a channel for polarization relaxation and presents an arguably even more serious problem. Due to the potentially large conducting volume and the tendency for impurity states to not localize in this class of compounds, this puts a very strong additional strong constraint on using a dc or low frequency ac techniques. It is important to point out that similar constraints likely apply to the proposals to induce a magnetic monopole image charge [73]. With current or foreseeable materials, experiments will likely have to be done with an oscillating cantilever to induce a transient image magnetic monopole. Pesin and MacDonald treated the related problem of the effective magnetic monopole induced near the surface of a TI in the presence of finite longitudinal conductance due to the presence of a suddenly introduced external electric charge [74]. In a very related fashion to the above they found that finite longitudinal conductivity introduces certain dynamical constraints on seeing the topological magnetoelectric effect.

# 7 Experiments

More or less simultaneously, three groups – performing experiment on bulk insulating $Bi_2Se_3$ [64] (by the present authors), Cr-doped $BiSbTe_3$ [76] and HgTe [61] – reported the observation of the topological magnetoelectric effect consistent with axion electrodynamics. In all these experiments, either a magnetic field was applied perpendicular to the film, or the film was uniformly magnetized putting these experiments in the regime of Fig. 8c and Figa. 10a, c, and d. Unless top and bottom surfaces are differentially doped to be electron and hole biased, in such a configuration the system is not expected to have a effective ME susceptibility (e.g. a polarization cannot be generated from an applied magnetic field), but it will still have a formal ME susceptibility that can be measured through its response to low frequency radiation. In these cases, researchers were looking to measure quantized Faraday and Kerr rotations accurately. This is similar to the original experiments proposed, although they were done in a slightly different fashion [77–79]. It is interesting to consider these experiments in light of the above discussion. Both the experiments on Cr-doped $BiSbTe_3$ [76] and HgTe [61] were performed on samples that were quite thin on the scale of the evanescent depth of the surface state wavefunctions and hence on the edge of the regime that should be considered 2D. This is

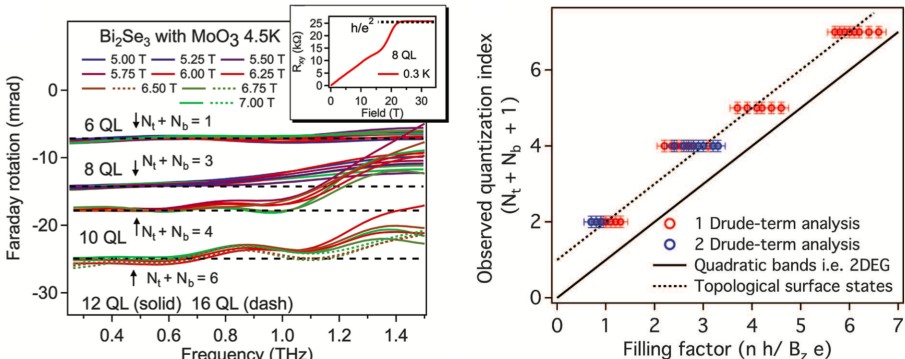

Figure 13: Left) Quantized Faraday rotation for different $Bi_2Se_3$ films. Dashed black lines are theoretical expectation values assuming certain values for the filling factor of the surface states. (inset) dc transport Hall resistance of a representative 8-QL sample. Right) Measured quantization index versus filling factor. The solid line is the expectation for quadratic bands, and the dashed line is for two topological surface states. From Ref. [64].

important in inferring the existence of an isolated 1/2 quantized surface Hall conductance. In the case of the magnetic doped TI the material does have inversion symmetry on average, but the experimental quantization is imprecise, presumably due to sample inhomogeneity coming from large amounts of magnetic dopants [80] and even at 1.5 K the surface states are not fully gapped at the chemical potential and therefore the sample is not completely in the quantum anomalous Hall regime [76]. In contrast, in the cleaner $Bi_2Se_3$ samples [81], an external magnetic field with a few Tesla is large enough to put the chemical potential in the fully gapped surface states [64]. The Faraday rotation on $Bi_2Se_3$ is quantized as shown in Fig 13. The value is given by

$$\tan(\phi_F) = \frac{2\alpha}{1+n}\left(N_t + \frac{1}{2} + N_b + \frac{1}{2}\right), \tag{22}$$

where $n$ is the refractive index of the substrate and $N_t$ and $N_b$ are Landau Level index of the top and bottom surfaces. As discussed in Ref. [64, 76], measuring Faraday and Kerr rotation at the same time can probe the quantization and give a direct measure of the fine structure constant $\alpha$. $\alpha$ was measured on a macroscopic $Bi_2Se_3$ sample to within 0.5% error [64]. In Fig. 13, a plot of observed quantized index vs. filling factor is a direct evidence that one observed the contribution of two topological surface states, each of which contributes to half-integer quantum Hall conductance and therefore provide the evidence for the topological magneto-electric effect [64]. Recent work utilizing ionic liquid gating successfully tuned the chemical potential as low as 10 meV above the Dirac point and pushed the sample across a few surface quantum Hall plateaus [75], as shown in Fig.14. This experiment is a direct measure of the formal ME susceptibility and the ME susceptibility lattice.

All of these experiments are manifestations of what was called the quantum Faraday effect in Ref. [24], which is again a configuration where the axion angle is an increasing function of space as in Fig. 10a or c. There has been no experimental measure of quantization in the regime shown in Fig. 10b that could be expected to manifest a true ME with finite effective ME susceptibility for the reasons discussed above. However, as is hopefully clear from this discussion there is no intrinsic difference from one scenario the other. They are all just different demonstrations of the same underlying physics and both experiments are measures of the formal ME susceptibility.

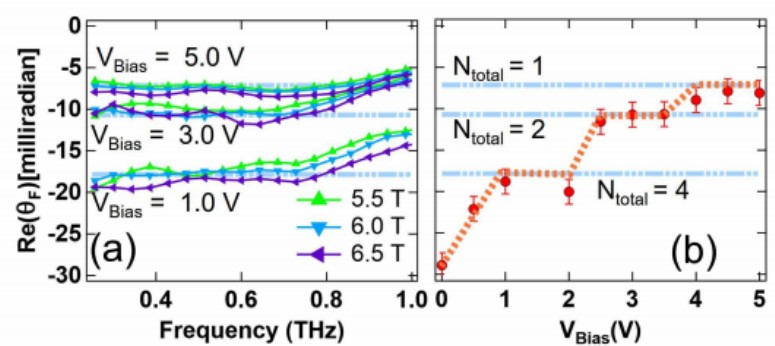

Figure 14: Left) (a) Real part of Faraday rotation ($\theta_F$) at high magnetic field for a sample with ionic liquid gated at difference voltages, B > 5.5T. The grey lines are theoretically predicted values assuming particular filling factors of the surface states. (b) Average value of Re $\theta_F$ over frequency range spanning from 0.2 to 0.8 THz at 6.5 T at different values of the bias voltage ($V_{Bias}$) From Ref. [75].

The interpretation of all these experiments in terms of quantized formal ME response rests on the fact that symmetry constrains the bulk axion angle to be an odd integer times $\pi$ in the bulk. Therefore the observation of a quantum Hall odd integer sum of top and bottom surfaces can be interpreted as a half-integer quantum Hall effect of a single surface. However, it is still desirable to isolate the half-integer Hall conductance of a single surface. It may be possible to do this through performing a THz reflection experiment off of a single surface directly. This would be a completely model free measurement of the formal ME lattice in much the same fashion as the measurement of a single end charge establishes the formal polarization lattice of a 1D chain. Although in principle possible, such an experiment has not yet been performed. It will require thick single insulating crystals.

## 8   Concluding remarks

This article is an attempt to explain in plain language how and why topological insulators should be regarded as magnetoelectric materials. We have drawn inspiration from the related example of electrical polarization in 1D and the concepts of formal vs. effective polarization. In so doing we gain important insight on the formal vs. effective magnetoelectric susceptibility, the $\frac{1}{2}$ quantized surface quantum Hall effect, the role of inversion symmetry, and the role of finite frequency measurements. Going forward one potentially fascinating, but as of yet unrealized related state of matter is that of the "intrinsic axion insulator". These are theoretically proposed [82, 83] stochiometric materials with a large ME response that originates in the same Chern-Simons contribution to the ME tensor that gives topological insulators their ME response. Roughly speaking these will be band-inverted materials "like" topological insulators, but with extant magnetism; the TI surface states are then intrinsically gapped such that a bulk sample exhibits a large intrinsic ME response. Related materials have been found where the magnetism is achieved by doping, but what is desired is a pure material that shows magnetism in this fashion. A number of compounds were theoretically proposed some years ago in Refs. [82,83] and much more recently in Ref. [84]. The material proposed in the latter work $MnBi_2Te_4$ has been synthesized and shown to be magnetic and possess topological surface states [85,86], but the very large parasitic bulk conductances will destroy any magne-

toelectric effect. In the event that bulk insulating samples can be synthesized the discussion in this manuscript will directly apply.

## Acknowledgements

We would like to thank A. Burkov, F.W. Hehl, T. Hughes, J. Moore, A. Pimenov, A. Rosch, A. Turner, A. Vishwanath, and particularly D. Vanderbilt for key conversations or correspondences. We would also like to thank Seongshik Oh for continuing collaboration and his generosity with providing his thin films – the world's best $Bi_2Se_3$. Without the advances he has pioneered none of our experiments would have been possible. NPA would also like to sincerely thank V. Yakovenko for an argument that got him started thinking in-depth on these issues and the hospitality of the Institute for Solid State Physics (ISSP) at the University of Tokyo where this manuscript was finally finished.

**Funding information**   Work at JHU on these topics is currently supported by the Army Research Office under Grant W911NF-15-1-0560 and the NSF EFRI 2-DARE program Grant No. 1542798. LWs work at UP on this topic is partially supported by a seed grant from the UP's NSF supported Materials Research Science and Engineering Center (MRSEC) (DMR-1720530).

## A   Derivation of "modified" Maxwell's Equation

One may derive the "axion modifications" to the Maxwell equations appearing in Eqs. 3 and 4 using a modified version of the standard Langrangian treatment, where it is mandated that the action be stationary with respect to variations of the potentials. The two terms in the Lagrangian density in potential form are

$$\mathcal{L}_0 = \frac{\epsilon_0}{2}\left(\nabla\phi + \frac{\partial \mathbf{A}}{\partial t}\right)^2 - \frac{1}{2\mu_0}(\nabla\times\mathbf{A})^2 - \rho\phi + \mathbf{J}\cdot\mathbf{A}, \tag{23}$$

$$\mathcal{L}_\theta = 2\alpha\sqrt{\frac{\epsilon_0}{\mu_0}}\frac{\theta}{2\pi}\left(\nabla\phi + \frac{\partial \mathbf{A}}{\partial t}\right)\cdot(\nabla\times\mathbf{A}). \tag{24}$$

They represent the conventional (Maxwell) and topological (axion) contributions to the Lagrangian respectively. Here $\alpha$ is the fine structure constant and $\epsilon_0$ and $\mu_0$ are the permittivity and permeability of the free space. The topological contribution is distinguished by $\theta$, which is an angle that will assume different values inside and outside the material of interest.

If either $\mathcal{T}$ or $\mathcal{P}$ symmetry is present, its value is quantized to be an even or odd integer times $\pi$ modulo $2\pi$. The action is

$$\mathcal{S} = \mathcal{S}_0 + \mathcal{S}_\theta = \int dt\, d^3x\,(\mathcal{L}_0 + \mathcal{L}_\theta), \tag{25}$$

where $\mathcal{S}_\theta$ derives from the additional term and $\mathcal{S}_0$ is the usual Maxwell action. One can start with with Eq. 25 and perform the typical variation of the potentials in the action to get modifications to Gauss's law and Ampère's law. The modified Gauss's law term comes from variations in the scalar potential $\phi$. One defines

$$\delta\mathcal{S} = \mathcal{S}(\phi + \delta\phi) - \mathcal{S}(\phi) = \delta\mathcal{S}_0 + \delta\mathcal{S}_\theta, \tag{26}$$

where $\delta\phi$ is an infinitesimal. As found in standard references [87] the Maxwell part of the variation can be written as

$$\delta\mathcal{S}_0 = -\int dt\, d^3x \left[ \epsilon_0 \nabla \cdot \left( \nabla\phi + \frac{\partial\mathbf{A}}{\partial t} \right) + \rho \right] \delta\phi. \tag{27}$$

For the new term, to first order in $\delta\phi$ one has the variation

$$\delta\mathcal{S}_\theta = -\int dt\, d^3x \left[ 2\alpha\sqrt{\frac{\epsilon_0}{\mu_0}} \frac{\theta}{2\pi} (\nabla\times\mathbf{A})\cdot\nabla\delta\phi \right]. \tag{28}$$

As with the Maxwell term, one shifts the derivatives to the other spatially dependent terms in the integrand by integration by parts. The surface terms can be set to zero. One has

$$\delta\mathcal{S}_\theta = \int dt\, d^3x\, \nabla \cdot \left[ 2\alpha\sqrt{\frac{\epsilon_0}{\mu_0}} \frac{\theta}{2\pi} (\nabla\times\mathbf{A}) \right] \delta\phi. \tag{29}$$

Expanding the divergence and using the fact that the divergence of a curl is zero one has

$$\delta\mathcal{S}_\theta = \int dt\, d^3x \left[ \nabla\left(\frac{\theta}{2\pi}\right) \cdot 2\alpha\sqrt{\frac{\epsilon_0}{\mu_0}} (\nabla\times\mathbf{A}) \right] \delta\phi. \tag{30}$$

We add this to the variation of the usual Maxwell action to get

$$\delta\mathcal{S} = \int dt\, d^3x \left[ -\left( \epsilon_0\nabla\cdot(\nabla\phi + \frac{\partial\mathbf{A}}{\partial t}) + \rho \right) + 2\alpha\sqrt{\frac{\epsilon_0}{\mu_0}} \nabla(\frac{\theta}{2\pi})\cdot(\nabla\times\mathbf{A}) \right] \delta\phi. \tag{31}$$

Setting the variation of this total action to zero requires the term in the brackets be equal to zero. Rearranging and substituting back in for the fields, one gets a modified version of Gauss's law (Eq. 3) with the additional source term that we gave above.

To get the modified version of Ampère's law one must vary the vector potential. Expanding $\mathcal{S}(\mathbf{A}+\delta\mathbf{A})$ to first order in $\delta\mathbf{A}$ one has for the Maxwell term

$$\delta\mathcal{S}_0 = \int dt\, d^3x \left[ -\epsilon_0 \frac{\partial(\nabla\phi + \partial\mathbf{A}/\partial t)}{\partial t} - \nabla\times(\nabla\times\mathbf{A})/\mu_0 + \mathbf{J} \right] \cdot \delta\mathbf{A}. \tag{32}$$

For $\mathcal{S}_\theta$ we have

$$\delta\mathcal{S}_\theta = \int dt\, d^3x \left[ 2\alpha\sqrt{\frac{\epsilon_0}{\mu_0}} \frac{\theta}{2\pi} \left( \frac{\partial\delta\mathbf{A}}{\partial t}\cdot(\nabla\times\mathbf{A}) + \left(\nabla\phi + \frac{\delta\mathbf{A}}{\delta t}\right)\cdot(\nabla\times\delta\mathbf{A}) \right) \right]. \tag{33}$$

Now we integrate by parts by moving the derivative with respect to time on the first term and the gradient on the second. Setting the surface terms to zero and after some simplification one gets

$$\delta\mathcal{S}_\theta = \int dt\, d^3x \left[ 2\alpha\sqrt{\frac{\epsilon_0}{\mu_0}} \left( \frac{\partial\theta/\partial t}{2\pi}(\nabla\times\mathbf{A}) - \nabla\left(\frac{\theta}{2\pi}\right)\times\left(\nabla\phi + \frac{\partial\mathbf{A}}{\partial t}\right) \right) \right] \cdot \delta\mathbf{A}. \tag{34}$$

The total variation with respect to the vector potential then reads

$$\delta\mathcal{S} = \int dt\, d^3x \left[ -\epsilon_0\frac{\partial(\nabla\phi + \partial\mathbf{A}/\partial t)}{\partial t} - \nabla\times(\nabla\times\mathbf{A})/\mu_0 + \mathbf{J} \right.$$
$$\left. + 2\alpha\sqrt{\frac{\epsilon_0}{\mu_0}} \left( \frac{\partial\theta/\partial t}{2\pi}(\nabla\times\mathbf{A}) - \nabla\left(\frac{\theta}{2\pi}\right)\times\left(\nabla\phi + \frac{\partial\mathbf{A}}{\partial t}\right) \right) \right] \cdot \delta\mathbf{A}. \tag{35}$$

As before if the total variation is to be zero for any infinitesimal $\delta \mathbf{A}$ then the quantity in brackets must be zero. Rearranging and again substituting in for the fields, one finds the modified version of Ampère's law with an additional current term that we we have above in the main text (Eq. 4).

Again it should be noted that it more conventional treatments of magnetoelectrics [9, 10, 17] the magnetoelectric properties are introduced into the constitutive equations for the material itself and not directly into the Maxwell's equations. Ours is an effective description which is largely equivalent, but Maxwell's equations are not "really modified" as they are fundamental laws based on electric charge and magnetic flux conservation. We use this description here for historical reasons [15] and the fact that it allows a direct perspective on how surface properties are modified by the axion physics in for instance the pumping of axion coupling.

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
