# Peer review of "On the matter of topological insulators as magnetoelectrics"

_SciPost Physics, doi:SciPost Phys. 6, 046 (2019)_

## Round 1 · Referee Report · Anonymous (Referee 1) · 2018-11-14

Report

I have read through the manuscript by Wu and Armitage that was sent to me with great interest. The paper addresses the tricky question of the relationship of the magnetoelectric (ME) effect and the properties of topological insulators. The former has been studied for decades with all researchers agreeing with the fundamental argument that materials exhibiting ME effects must break both inversion (P) and time-reversal (T) symmetries. Nevertheless, a by-product of the recent breakthroughs in the area of topological insulators is the prediction that they also exhibit ME effects, despite not necessarily breaking either symmetry. Wu and Armitage discuss this seeming paradox in great detail and at a level that addresses a wide audience. As a result I wholeheartedly recommend the endorsement of this paper by SciPost.
  • validity: -
  • significance: -
  • originality: -
  • clarity: -
  • formatting: -
  • grammar: -

Author:  N. Peter Armitage  on 2018-11-26  [id 359]

(in reply to Report 2 on 2018-11-14)

We appreciate the positive comments of the referee

Author:  N. Peter Armitage  on 2018-11-26  [id 358]

(in reply to Report 2 on 2018-11-14)

We appreciate the positive comments of the referee.

---

## Round 1 · Referee Report · Anonymous (Referee 2) · 2018-11-19

Strengths

1) The paper has an accessible discussion to a broad audience. 2) The paper discusses at length various conceptual subtleties surrounding the magneto-electric effect in 3D topological insulators.

Weaknesses

1) The paper illuminates on the discussion of the subtleties of the quantised magneto-electric effect in 3D topological insulators. However, at the same time, it tries to advocate the view, somewhat in between the lines, that a measurement of this effect is not that significant or important. Also it seems to bypass the question on whether the single isolated 1/2 quantised Hall conductivity in the surface of the TI can be measured.

Report

The paper is a valuable addition to the discussion of the subtleties surrounding the magneto-electric effect in topological insulators. I found the discussion very instructive and at a level that can help communicate these issues to a broad audience.

In spite of the conceptual value of their discussion, the authors seem to suggest that there is not much value in pursuing the measurement of the quantised magneto-electric effect, and that this is basically a closed matter given that they have already measured the quantised Faraday rotation. There is no doubt that this effect is intimately related to the Faraday rotation effect described by the authors. But even though they are intimately related it is healthy not to blur completely their identities (think of the zero resistance and the perfect diamagnetism of a superconductor which are also intimately related to each other and come hand-in-hand in the superconducting state yet it is important to distinguish them).

More importantly, in my opinion no measurement to this date has been able to detect the isolated 1/2 quantised conductance expected at the surface of the TI (under the right symmetry breaking conditions to gap the surfaces). This is the essentially anomalous feature of a single surface of the 3D TI (which cannot be mimicked by any stand alone 2D band insulator). All measurements to this date basically contain additive contributions of the two surfaces making the result strictly non-anomalous. This a key aspect of the discussion that seems to have been mostly overlooked in the paper.

Requested changes

1) In Eq(11) they mean "Re" instead of "Im"?

2) The argument that leads to Eq.(16) overlooks one important fact: that threading one flux per surface unit cell is an extremely large perturbation, so adiabaticity is far from guaranteed. In fact it is known that when half-flux quantum is threaded per unit cell strong topological insulators develop a kind of 1D metallic wire along the flux tube, in the form of a pair of gapless conter-propagating 1D gapless modes that penetrates into the bulk (see e.g. Phys. Rev. B 82, 041104(R) (2010)). Perhaps a safer argument can be made by assuming an enlarged unit cell. Can the authors should clarify or remove these arguments?

3) The authors state: "An effective magnetoelectric susceptibility can only be de?fined in ... where the net Hall response is zero." (notice missing word ...="systems"). This is an important point, and I kind of see why (my view: if it is non-zero the system might have net charge accumulation from Streda formula after threading magnetic field, also surface cannot be fully insulating and hence charge might flow). But the authors should explain why this makes the effective magnetoelectric susceptibility ill defined.

4) The authors state: "The hybrid Wannier function representation makes explicit the fact that one cannot create Wannier functions in such a topological systems despite the fact that the eigenstates of Hamiltonian have the Bloch form." I guess the authors mean that one cannot create Wannier functions that are localized and strictly respect the symmetry? (see e.g. Phys. Rev. B 83, 035108 (2011), Phys. Rev. B 93, 035453 (2016)).

5) Related limitations to the measurement of the apparent monopole at the surface of a TI described here were also discussed in Phys. Rev. Lett. 111, 016801 (2013).

6) The authors write: "Again by way of analogy with the 1D chain, this suggests a way of looking at inversion symmetric insulators as overlapping e^2/h and -e^2 /h layers. As shown in Fig. 8, one can conceive of conventional insulators as being materials these conducting layers are centered on top of each other and spatially overlap and cancel, whereas a TI is where layers of them are displaced from each other by half a unit cell, giving 1/2e^2/h on the surface." How literal should this picture be taken? e.g. how is time reversal supposed to act in this hypothetical system of displaced quantum Hall layers? or, are the authors then imagining a system with large breaking of TRS throughout the bulk? if so, how to think then about TR invariant TI's?

7) In view of the comments in report above, the authors might want to revisit/rephrase statements such as:

"Although the development of systems that realize this con?guration is very important from a materials perspective, we do not believe it warrants any particular consideration as anything special or fundamental. Both scenarios have the same formal ME susceptibility. As shown in Fig. 10, the two con?figurations should just be considered as di?fferent experimental conditions and realize fundamentally the same thing."

"However, as is hopefully clear from this discussion there is no intrinsic di?erence from one scenario the other. They are all just di?fferent demonstrations of the same underlying physics and both experiments are measures of the formal ME susceptibility."

  • validity: good
  • significance: good
  • originality: good
  • clarity: good
  • formatting: good
  • grammar: good

Author:  N. Peter Armitage  on 2018-12-30  [id 395]

(in reply to Report 1 on 2018-11-19)
Category:
remark
answer to question
reply to objection

Please see attached pdf

Attachment:

RefereeReply.pdf

---

## Round 2 · Author Response

Dear Editor,

We have replied to all the comments and critiques of the referees and hope that you and they will now find the manuscript suitable for SciPost.

-Peter Armitage

---

## Round 2 · List of Changes

• In this revised version we have added some additional discussion on the significance of the effect we found. Moreover near the end of the manuscript we have added some text on how the isolated 1⁄2 quantized response can be measured directly.

  • We have added text explaining what experiments we believe still need to be done.

  • We now say so explicitly that no experiment has measured an isolated single surface.

  • We have changed Im to Re for Eq. 11

  • We cite additional papers that the referee has pointed out on the fact that one cannot create Wannier functions that are localized and strictly respect the symmetry?

  • We have added discussion on the important paper of Pesin and MacDonald.

  • We have also now added a table at the end of the manuscript that makes a comparison between 1D polarization and the 3D ME effect.

---

## Editorial Decision

published